# Distilled Thompson Sampling: Practical and Efficient Thompson Sampling via Imitation Learning

**Hongseok Namkoong**                                  *namkoong@gsb.columbia.edu*
*Decision, Risk, and Operations Division*
*Columbia Business School*

**Samuel Daulton**                                  *sdaulton@meta.com*
*Meta*

**Eytan Bakshy**                                  *sdaulton@meta.com*
*Meta*

**Reviewed on OpenReview:** *https://openreview.net/forum?id=J8PrWwvYX2*

## Abstract

Thompson sampling (TS) has emerged as a robust technique for contextual bandit problems. However, TS requires posterior inference and optimization for action generation, prohibiting its use in many online platforms where latency and ease of deployment are of concern. We operationalize TS by proposing a novel imitation-learning-based algorithm that distills a TS policy into an explicit policy representation, allowing fast decision-making and easy deployment in mobile and server-based environments. Using batched data collected under the imitation policy, our algorithm iteratively performs offline updates to the TS policy, and learns a new explicit policy representation to imitate it. Empirically, our imitation policy achieves performance comparable to batch TS while allowing more than an order of magnitude reduction in decision-time latency. Buoyed by low latency and simplicity of implementation, our algorithm has been successfully deployed in multiple video upload systems for Meta. Using a randomized controlled trial, we show our algorithm resulted in significant improvements in video quality and watch time. We release our implementation as open-source software here.

## 1 Introduction

In the past decade, Thompson sampling (Thompson, 1933) has emerged as a powerful algorithm for contextual bandit problems. The underlying principle is simple: an action is chosen with probability proportional to it being optimal under the current posterior distribution. Driven by the algorithm's strong empirical performance (Scott, 2010; Chapelle & Li, 2011; May & Leslie, 2011), many authors have recently established rigorous performance guarantees (Kaufmann et al., 2012; Agrawal & Goyal, 2013a;b; Gopalan et al., 2014; Honda & Takemura, 2014; Russo & Van Roy, 2014; Abeille et al., 2017). Thompson sampling is increasingly being applied to a broad range of applications including revenue management (Ferreira et al., 2018), internet advertising (Graepel et al., 2010; Agarwal et al., 2014; Schwartz et al., 2017), and recommendation systems (Kawale et al., 2015).

Despite its conceptual simplicity and strong performance, Thompson sampling can be difficult to deploy in practice. Thompson sampling consists of two steps: *posterior sampling* and *optimization. Posterior sampling* requires evaluating a potentially large number of actions from a well-calibrated probabilistic model. Accurately calibrating uncertainty is important for optimally trading off exploration and exploitation, and is critical to practical performance (Riquelme et al., 2018). Large-scale probabilistic machine learning models based on deep networks show much promise as they can adaptively learn good feature representations for

uncertainty calibration (Wang & Yeung, 2020). However, sampling from these probabilistic models can be demanding in terms of computation and memory. While approximate inference methods with better runtime characteristics exist, they often produce poorly calibrated uncertainty estimates that lead to poorer empirical performance (Riquelme et al., 2018). The second step, *optimization*, solves for a reward-optimizing action under the posterior sample. This can also be prohibitively expensive when the action space is large or continuous. For example, an advertising platform that matches advertisers to users at each time period has to solve combinatorial optimization problems real-time in order to run Thompson sampling (Mas-Colell et al., 1995).

For typical online platforms, low latency—real-time computational performance—is critical for user satisfaction and retention. The *online* nature of the computation required for Thompson sampling thus poses a substantive challenge to deploying it in large-scale internet services. These challenges are especially pronounced in resource-constrained mobile applications, a ubiquitous modality for modern internet applications: as of 2018, an estimated 52.2% of worldwide web traffic was generated by mobile devices (Statistica, 2020). Mobile applications require decisions to be made in a fast and memory-efficient manner, and on-device decision-making is important to good user experience in domains such as adaptive video streaming (Mao et al., 2019) and social media ranking (Petrescu & Tas, 2016). However, the majority of internet-connected mobile devices have limited memory, and utilize low-end processors that are orders of magnitude slower than server-grade devices (Bhardwaj et al., 2019; Wu et al., 2019). As affordable, compute-limited mobile devices are increasingly adopted in developing economies (Ricciardi, 2019), the ability to deploy cutting-edge decision algorithms on diverse computing infrastructure is important for democratization of technology and long term business growth.

Software development cost is another core practical consideration when implementing contextual bandit algorithms in large-scale online platforms. Long-term software development cost is commonly referred to as tech debt, which is incurred when a suboptimal, myopic development plan is followed in lieu of one that requires (sometimes much) higher initial effort, but less future work. Avoiding tech debt is critical to a reliable and scalable service (Sculley et al., 2015; Ramasubbu & Kemerer, 2016; Banker et al., 2021), but contextual bandit systems are challenging due to their high complexity: they require temporal feedback loops consisting of different pipelines on exploration, data logging, policy updates, and deployment (Agarwal et al., 2016). The *online* nature of the complex numerical routines required by Thompson sampling significantly exacerbate these practical difficulties. *Real-time* posterior sampling and action optimization leads the overall system to be cumbersome and hard to debug, posing challenges to reliable software development.

**Example 1** (Video Transcoding)**:**  As our main real-world application, we study video uploads for large online platforms. Video is an increasingly popular medium on social networks, but uploading video is still a technically challenging problem, where limited bandwidth and compute capacity—particularly problematic on mobile devices—leads to unsuccessful uploads. When a user requests a video be uploaded to a social media service, the service must choose the desired video quality (bitrate) for transcoding the video before uploading. Video needs to be optimally transcoded considering quality, and success of file upload. It is preferable to upload videos at a high quality because it can lead to a better viewer experience (if the viewer has a sufficiently good network connection). However, higher quality videos have larger file sizes, making it more likely to fail to upload: larger files take longer time to upload, increasing the likelihood that the network connection to fail, or the user to grow frustrated and cancel the upload.

We are interested in an online platform who wish to make contextual decisions about how to optimally transcode a video at upload time. Making such decisions quickly is critical for user satisfaction; low latency is particularly important for popular short-form videos uploaded on Tik-Tok, Snapchat, and Instagram, where videos are captured and uploaded frequently and in real-time. Although transcoding decisions needs to be made quickly in order to be responsive and keep the user engaged, most upload requests come from resource-constrained mobile devices. ◇ ◇

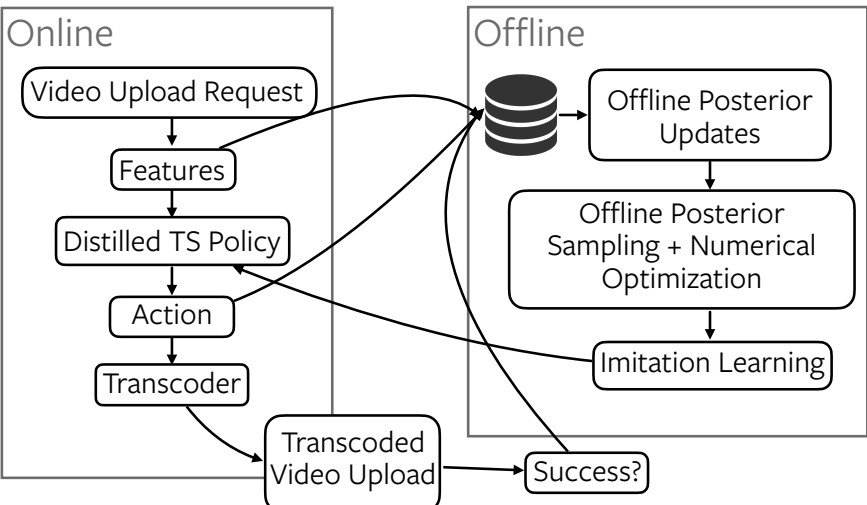

Figure 1: An illustration of distilled TS on the example video uploads application, described in Example 1. *Online* action generation is performed asynchronously on resource-constrained mobile edge devices whereas batched policy updates are performed *offline* on powerful backend servers.

The problem motivation goes beyond our main application. There are numerous examples of decision-making problems on online platforms where latency and system complexity are of central concern.

**Example 2** (Advertising on third party systems)**:** Every time a user arrives to a third party webpage (e.g. New York Times), the advertising platform (e.g. Google Ads) decides which ad to show in order to maximize conversion. Latency is important to good user experience (Agarwal et al., 2016), and curbing system complexity increases service reliability (Sculley et al., 2015). ⋄⋄ **Example 3** (Ranking)**:** When a

user logs in, an internet service chooses a list of items to display to the user in order to maximize revenue or engagement. Concrete examples include ranking news articles (Microsoft Network, MSN), products (online marketplaces like Amazon and Airbnb), and content (Facebook and LinkedIn feed). In all of these cases, latency is central to user satisfaction, but mobile edge devices and front-end servers are resource constrained (Agarwal et al., 2016). For instance, there has been work from Meta Facebook on performing secondary ranking on device to avoid server communication latency and to only display content that has been downloaded completely (Petrescu & Tas, 2016).⋄⋄

**Example 4** (Personalized Pricing)**:** As a customer enters a virtual platform, the system generates a personalized price based on market conditions and user-specific contexts. Electronic commerce firms and airlines use price controls to manage revenue (Talluri & Van Ryzin, 2004; Den Boer, 2015), and two-sided online marketplaces (e.g. Uber, Lyft, Airbnb) dynamically set prices on both sides of the market to reduce supply-demand imbalance. In both cases, latency is important for a satisfactory user experience. ⋄⋄

**Methodology** Motivated by aforementioned challenges in implementing and deploying Thompson sampling on online platforms, we develop and analyze a method that maintains an explicit policy representation designed to imitate Thompson sampling. In order to avoid computationally demanding routines *online*, our algorithm simulates and imitates a Thompson sampling policy *offline*. An explicit policy representation can efficiently generate actions real-time even in large action spaces, without requiring real-time posterior inference or numerical optimization. An illustration of how this methodology can be applied to video transcoding (Example 1) is provided in Figure 1. This allows leveraging state-of-the-art Bayesian models—such as Gaussian processes parameterized by deep neural networks—and optimization solvers *offline*, while maintaining low

latency on resource-constrained computing modalities such as low-end mobile devices.[1] During operation, actions can be generated efficiently from the distilled policy by sampling from a parameterized distribution, allowing fast and asynchronous interaction with users. For example, recent engineering progress allows generating actions using an industrial-scale neural network model in 0.3880 milliseconds (Coleman et al., 2017).

By performing posterior updates and mimicking the behavior of Thompson sampling offline, we are able to move complex numerical routines from resource-constrained mobile devices to backend servers, and reduce long-term software development costs (tech debt). Such offline procedures using batched observations can be easily implemented using modern industry machine learning pipelines (Gauci et al., 2018; Fujimoto et al., 2019). This allows leveraging the recent remarkable progress in machine learning software infrastructure, such as engineering best practices and tools for reliable testing & deployment[2].

Our approach shifts computation from online latency-critical to offline batch processing environments, and does not produce a universally dominating intervention. This trade-off is most useful when offline computation can be carried out on powerful backend servers for a relatively extended period of time, whereas online computation must execute instantaneously on resource-constrained devices with strict latency requirements.

**Practical impact**  Empirically, we evaluate our imitation algorithm on several benchmark problems and a real-world dataset for selecting optimal video transcoding configurations (Section 4). In all of our experiments, our imitation algorithm performs as well as batch on-policy Thompson sampling in terms of cumulative regret, while reducing decision-time latency by an order of magnitude. Buoyed by low latency and simplicity of implementation showcased in our empirical benchmarking efforts, our imitation learning policies have been used in video upload systems across Meta products, which are leading social networking services. Our contextual policy tunes the bitrates for video uploads based on contextual features such as download bandwidth, device model, operating system, connection class (2G, 3G, 4G), country, and video features which include source resolution, bitrate, and file size.

To assess the impact of our algorithm, we ran internal randomized controlled trials (RCT) on each of the aforementioned products. We find our algorithm achieves significant improvements in video quality, which we measure using the fraction of videos with quality preserved at 1080p (high resolution). Our RCTs show up to 5x improvements over existing video upload policies on all surfaces.

The RCTs show significant increases in topline metrics that are of importance at the company level. Due to better video quality, we observed increased video watch times on multiple products: 1.1% on Facebook iOS Feed videos, 0.77% on Facebook Android Feed videos, 0.27% on Facebook Android Stories, 0.45% on Instagram Stories. In addition, our contextual policies boosted interaction metrics on several products: increases in meaningful social interactions of 0.15% and 0.14% on Facebook Android Stories and Facebook Android Feed, respectively, and an increase in interactions of 0.26% on Instagram Stories. All findings were significant at the 95% level.

Buoyed by these results, our contextual policy has been deployed across multiple product verticals including Facebook Feed, Stories, Reels and Instagram Stories and Reels. Our algorithm has been independently applied to both iOS and Android apps for all aforementioned products, and is reliably handling millions of uploads each day.

**Theoretical contributions**  To understand the strong practical advantages we showcase, we take initial steps toward a principled understanding of our imitation algorithm. Since our (batch) updates to the Thompson sampling policy are based on observations generated by the imitation policy, our algorithm emulates an *off-policy* version of Thompson sampling which may diverge from its on-policy counterpart. Due to its off-policy nature, an uninformed and pessimistic view of our procedure states that any initially small

---

[1]More generally, optimization can be a challenge for non-Bayesian methods. Although outside of the scope of this paper, generalizing our imitation framework to other policies will likely yield fruit in separating optimization from online action-generation.

[2]As an example, a dedicated top peer-reviewed conference for ML systems https://mlsys.org/ was recently established, and is undergoing rapid growth at the forefront of academia and industry. This community focuses on improving the efficiency of ML systems from an *operational* perspective.

deviation between the imitation policy and Thompson sampling may cascade across time. Our main theoretical results (Section 6-5) preclude such possibility and ensure small deviations between the imitation policy and Thompson sampling do not magnify over time. Specifically, we show that our imitation policy enjoys Bayes regret similar to that of batch *on-policy* Thompson sampling, up to the sum of single-step imitation errors. We substantiate our performance guarantees in general modeling scenarios involving contextual Gaussian processes, where a cleverly initialized version of our algorithm (albeit impractical) achieves advantageous Bayes regret (Section 5.2).

Solving the imitation problem, or equivalently, finding the policy parameterization closest to Thompson sampling, only requires unsupervised contexts—those without corresponding actions or rewards. On large-scale online platforms, unsupervised contexts are typically cheap and abundant, e.g., the entire user database provides a wealth of such contexts. In Section 6, we prove that each single-period imitation error term can be controlled—with a sufficiently rich imitation model—at the rate $O_p(1/\sqrt{N})$, where $N$ is the number of supervised and unsupervised contexts. Combining this with our aforementioned regret bound in Section 5, our imitation algorithm achieves Bayes regret comparable to batch on-policy Thompson sampling up to $O(T\sqrt{\log T}/\sqrt{N})$-error, where $T$ is the number of batched policy updates.

Despite the seemingly linear gap in Bayes regret, $N$ is typically orders of magnitude larger than $T$ in internet applications where we can utilize the database of users / entities. Typically, $N$ is in the order of hundreds of millions; as of 2020, Facebook had 2.7 billion monthly active users; in our motivating video transcoding application, the service receives millions of video upload requests *every day*, providing an effectively unlimited number of unsupervised contexts. In contrast, the number of model updates (horizon $T$) is relatively small, in hundreds, due to complexities of policy deployment and nonstationary user behavior. In such practical problem instances, our imitation policy thus enjoys Bayes regret bounds comparable to that of batch on-policy Thompson sampling.

**Paper Organization**   Section 2 reviews related work on Thompson sampling, approximate inference, and imitation learning. Section 3 formally introduces our distilled Thompson sampling algorithm. Section 4 presents empirical evaluations on benchmark problems and our real-world video transcoding application, demonstrating both regret performance and latency improvements. Sections 5 and 6 provide theoretical analysis: Section 5 shows that imitation error does not accumulate over time and relates our algorithm's Bayes regret to batch UCB algorithms, while Section 6 establishes generalization guarantees for the imitation learning objective. Section 7 concludes with discussion and future directions.

## 2   Related work

There is a substantial body of work on Thompson sampling and its variants that use computationally efficient subroutines. We give a necessarily abridged overview of how our algorithm situates with respect to the extensive literature on bandits, approximate inference, and imitation learning.

A number of authors have showed that Thompson sampling achieves optimal regret for multi-armed bandits (Agrawal & Goyal, 2012; 2013a; Kaufmann et al., 2012; Honda & Takemura, 2014). We refer the reader to the recent tutorial by Russo et al. (2018) and references therein for a comprehensive overview. Agrawal & Goyal (2013b); Abeille et al. (2017) showed regret bounds for linear stochastic contextual bandits for a Thompson sampling algorithm with an uninformative Gaussian prior, and Gopalan et al. (2014) studied finite parameter spaces. Russo & Van Roy (2014) established Bayesian regret bounds for Thompson sampling with varying action sets (which includes, in particular, contextual bandits); Russo & Van Roy (2016) provides an information-theoretic analysis that makes explicit the dependence on the prior (see also Bubeck & Eldan (2016)). We build on the insights of Russo & Van Roy (2014), and show that our imitation algorithm retains the advantageous properties of batch Thompson sampling, achieving (gap-independent) Bayes regret comparable to the *best* batch UCB algorithm.

Practical performance of Thompson sampling depends on having access to well-calibrated probabilistic predictions. Obtaining a balance between predictive accuracy, computational time, and memory requirements can be challenging in the context of large datasets with overparameterized models. Exact posterior sampling from even the simplest Gaussian linear models has a time complexity of $O(n^2)$, where $n$ is the number of

model parameters[3]. A common strategy used by some variational inference methods is to use a mean-field approach where parameters are assumed to be independent (Blundell et al., 2015). This assumption can decrease sampling costs from $O(n^2)$ to $O(n)$, where $n$ is the number of parameters. However, Riquelme et al. (2018) found that batch Thompson sampling using such approaches often leads to poor empirical performance.

When exact posterior inference is not possible, approximate inference methods can be used for posterior sampling. We refer the reader to Chapter 5 of Russo et al. (2018)'s recent tutorial for a discussion of approximation methods in relation to Thompson sampling. Bootstrapping (Eckles & Kaptein, 2014; Osband et al., 2016; Lu & Van Roy, 2017) is a simple heuristic procedure that maintains multiple models to approximate samples from the posterior distribution, although maintaining multiple models is often computationally expensive. MCMC-based methods for approximate inference, and Hamilton Monte Carlo (HMC) (Neal, 2011) in particular, are largely regarded as the "gold standard" for approximate Bayesian inference. HMC, and other MCMC-like approaches (e.g., Chen et al. (2014); Welling & Teh (2011)) generate an arbitrary number of posterior samples for all parameters. While such algorithms permit rapid evaluation of posterior samples (since the parameters are already sampled), they require substantial memory to store multiple samples of the parameters. Recent methods have also considered decomposing the covariance or precision matrix into a diagonal and low-rank component (Zhang et al., 2018; Maddox et al., 2019). While this reduces computational complexity and memory costs relative to using the full covariance, sampling still incurs a time complexity of $O((n+1)\rho)$ where $\rho$ is the rank of the covariance (or precision matrix) and $\rho$ copies of the weights must be stored.

By pre-computing and distilling Thompson sampling, our imitation learning framework allows the use of the most appropriate inferential procedure for the task at hand, rather than what is feasible to run in an online setting. In particular, the separation of online decision-making and offline computation allows the use of state-of-the-art Bayesian methods, such as those utilizing deep neural networks (Wang & Yeung, 2020). While we restrict discussion to Thompson sampling in this work, the basic idea of offline imitation learning can be used to learn a explicit policy representation of any complicated policy and allow operationalization at scale.

Imitation learning methods have received much attention recently, owing to their ability to learn complicated policies from expert demonstrations (Abbeel & Ng, 2004; Ross & Bagnell, 2010; Ho & Ermon, 2016). Our approach of minimizing the discrepancy between a parameterized policy and Thompson sampling can be viewed as an implementation of behavioral cloning (Ross & Bagnell, 2010; Syed & Schapire, 2010; Ross et al., 2011). Our imitation learning procedure resembles the "Bayesian dark knowledge" approach from Korattikara et al. (2015), which uses a neural network to approximate Bayesian posterior distributions. While most works in the imitation learning literature study reinforcement learning problems, we focus on the more limited contextual bandit setting, which allows us to show strong theoretical guarantees. We anticipate the growing list of works on imitation learning to be important in generalizing our imitation framework to the reinforcement learning (RL) setting. To account for time dependencies in state evolutions, both inverse RL approaches that directly model the reward (Abbeel & Ng, 2004; Syed & Schapire, 2008), and the recent advances in generative adversarial imitation learning techniques (Ho & Ermon, 2016; Li et al., 2017) show promise in generalizing our imitation algorithm (behavioral cloning) to RL problems.

## 3   Distilled Thompson sampling

Reflecting typical operational scenarios on online platforms, we consider a *batch* (Bayesian) contextual bandit problem. The agent / decision-maker generates actions *real-time* as user requests come in asynchronously, and performs batched, infrequent updates to the policy. In what follows, we formally introduce an imitation algorithm that makes it trivial to parallelize action generation over multiple computing nodes, even on each user's mobile device.

Let $\Theta$ be the parameter space, and let $\theta \sim P$ be a prior distribution on $\Theta$. At each time $t$, the agent observes a context, takes an action, and receives a reward: we denote the context $S_t \overset{\text{iid}}{\sim} \mathbb{P}_S$, action $A_t \in \mathcal{A}$, and reward $R_t \in \mathbb{R}$. We consider a well-specified reward model class $\{f_\theta : \mathcal{A} \times \mathcal{S} \to \mathbb{R} \mid \theta \in \Theta\}$

$$f_\theta(a, s) = \mathbb{E}[R_t \mid \theta, A_t = a, S_t = s] \text{ for all } a \in \mathcal{A}, s \in \mathcal{S}.$$

---

[3]This assumes the root decomposition of the covariance matrix has been cached, which incurs a cost of $O(n^3)$.

Let $H_t = (S_1, A_1, R_1, \ldots, S_{t-1}, A_{t-1}, R_{t-1})$ be the history of observations until time $t$. Assume that regardless of $H_{t'}$ for $t' \leq t$, the mean reward at time $t$ is determined only by the context-action pair

$$\mathbb{E}[R_t \mid \theta, H_{t'}, S_t = s, A_t = s] = f_\theta(a, s),$$

or equivalently, $R_t = f_\theta(A_t, S_t) + \epsilon_t$ where $\epsilon_t$ is a mean zero i.i.d. noise.

At time $t$, we denote by $\gamma(t)$ the period before which the most recent policy update occurred. For example, for a fixed batch size $B$

$$\gamma(t) = \begin{cases} 1 & \text{if } t = 1, \ldots B, \\ B+1 & \text{if } t = B+1, \ldots, 2B, \\ 2B+1 & \text{if } t = 2B+1, \ldots, 3B, \\ \vdots \end{cases} \tag{1}$$

More generally, we allow time-varying batch sizes that are a priori unknown to the decision maker. We use $\pi_{\gamma(t)}$ to denote the policy used at time $t$ that generates action $A_t$ based on the history $H_{\gamma(t)}$ available at the previous model update $\gamma(t)$: conditional on the history $H_{\gamma(t)}$, we have $A_t \mid S_t \sim \pi_{\gamma(t)}(\cdot \mid S_t)$, where we abuse notation to suppress the dependence of $\pi_{\gamma(t)}$ on the history $H_{\gamma(t)}$. In the sequential (non-batch) setting, we simply have $\gamma(t) = t$.

The agent's objective is to maximize the cumulative sum of rewards by updating the policy $\pi_{\gamma(t)}$ based on batches of context-action-reward observations. The *regret* of the agent compares the agent's cumulative reward to the reward under the optimal action: for any fixed parameter value $\theta \in \Theta$, the (frequentist) regret for the set of policies $\{\pi_{\gamma(t)}\}_{t \in \mathbb{N}}$ is

$$\text{Regret}\left(T, \{\pi_{\gamma(t)}\}_{t \in \mathbb{N}}, \theta\right) := \sum_{t=1}^{T} \mathbb{E}\left[\max_{a \in \mathcal{A}} f_\theta(a, S_t) - f_\theta(A_t, S_t) \mid \theta\right].$$

For simplicity, we assume $\arg\max_{a \in \mathcal{A}} f_\theta(a, s)$ is nonempty almost surely. We assume the agent's prior, $P$, is *well-specified*[4], a key (standard) assumption that drives our subsequent analysis. Under the prior $P$ over $\theta \in \Theta$, the Bayes regret is simply the frequentist regret averaged over $\theta \sim P$

$$\text{BayesRegret}\left(T, \{\pi_{\gamma(t)}\}_{t \in \mathbb{N}}\right) := \mathbb{E}_{\theta \sim P}[\text{Regret}\left(T, \{\pi_{\gamma(t)}\}_{t \in \mathbb{N}}, \theta\right)] = \sum_{t=1}^{T} \mathbb{E}_{\theta \sim P}\left[\max_{a \in \mathcal{A}} f_\theta(a, S_t) - f_\theta(A_t, S_t)\right].$$

Based on the history $H_{\gamma(t)}$, batch Thompson sampling plays an action according to the posterior probability of the action being optimal. The posterior probabilities are computed based on the prior $P$ and previously observed context-action-reward tuples. At time $t$, this is often implemented by

$$\text{sampling from the posterior } \theta_t \sim P(\theta \in \cdot \mid H_{\gamma(t)}, S_t) \text{ and solving } \bar{A}_t \in \arg\max_{a \in \mathcal{A}} f_{\theta_t}(a, S_t).$$

By definition, Thompson sampling enjoys the optimality property $\bar{A}_t \mid H_{\gamma(t)}, S_t \stackrel{d}{=} A_t^\star \mid H_{\gamma(t)}, S_t$ where $A_t^\star \in \arg\max_{a \in \mathcal{A}} f_\theta(a, S_t)$ and $\theta$ is the true parameter drawn from the prior $P$. Throughout, we assume $\bar{A}_t \mid H_{\gamma(t)}, S_t$ is independent of all else.

---

[4]When the prior is misspecified so that the Thompson sampling policy uses $Q$ instead of $P$, we have the equivalence as noted by Russo & Van Roy (2014)

$$\mathbb{E}_{\theta \sim P}[\text{Regret}\left(T, \{\pi_{\gamma(t)}\}_{t \in \mathbb{N}}, \theta\right)] \leq \left\|\frac{dP}{dQ}\right\|_{L^\infty(\mathcal{X})} \mathbb{E}_{\theta \sim Q}[\text{Regret}\left(T, \{\pi_{\gamma(t)}\}_{t \in \mathbb{N}}, \theta\right)],$$

where $dP/dQ$ is the Radon-Nikodym derivative of $P$ with respect to $Q$. While misspecified priors can incur substantially higher regret (Liu & Li, 2016) in the worst-case, empirical evidence suggests Thompson sampling is a strong algorithm in practice (Scott, 2010; Granmo, 2010; Chapelle & Li, 2011; May & Leslie, 2011; Ferreira et al., 2018; Graepel et al., 2010; Agarwal et al., 2014; Kawale et al., 2015; Schwartz et al., 2017; Agarwal et al., 2016).

To address challenges in implementing Thompson sampling real-time, we develop an imitation learning algorithm that separates *online* action generation from computationally intensive steps like posterior sampling and optimization. Our algorithm maintains an explicit policy representation that emulates the batch (off-policy) Thompson sampling policy by simulating its actions *offline*. At decision time, the algorithm generates an action simply by sampling from the current policy representation, which is straightforward to implement and computationally efficient to run real-time. We summarize an idealized form of our method in Algorithm 1, where conditional on the history $H_{\gamma(t)}$ generated by the imitation policy

$$\bar{\pi}_{\gamma(t)}(a \mid s) \text{ is the } \textit{batch off-policy Thompson sampling policy at time } t. \tag{2}$$

This policy is different from the true, batch on-policy Thompson sampling since the imitation policy generates actions based on which rewards are observed. Nevertheless, we will show that our algorithm enjoys Bayes regret comparable to batch on-policy Thompson sampling.

At each time $t$, our algorithm observes a context $S_t$, and plays an action drawn from its explicit policy representation. Formally, we parameterize our policy $\pi^m(a \mid s)$ with a model class $m \in \mathcal{M}$. For example, $\mathcal{M}$ can be a neural network that takes as input a context and outputs a distribution over actions. We generate actions by sampling from the current policy $A_t \sim \pi^m_{\gamma(t)}(\cdot \mid S_t)$, which can be easily implemented to run with low latency on resource-constrained computing infrastructure such as mobile devices. The agent uses a batch of context-action-reward tuples to update its posterior on the parameter $\theta \in \Theta$ *offline*. Although this step requires posterior inference that may be too burdensome to run real-time, our method allows running it offline on a different computing node, so that it does not affect latency. Using the updated posterior $\theta_t \sim \mathbb{P}(\cdot \mid H_{\gamma(t)})$, the agent then simulates actions drawn by the Thompson sampling policy by computing the maximizer $\bar{A}_t(s) \in \arg\max_{a \in \mathcal{A}} f_{\theta_t}(a, s)$, for a range of values $s \in \mathcal{S}$. Using these simulated context-action pairs, we learn an explicit policy representation that *imitates* the observed actions of the Thompson sampling policy.

---

**Algorithm 1** Imitating Batch Thompson Sampling

---

1: Input: prior $P$ on parameter space $\Theta$, reward model class $\{f_\theta(\cdot, \cdot)\}$, imitation policy model class $\{\pi^m : m \in \mathcal{M}\}$, notion of distance $D$ for probabilities
2: Initialize $m \leftarrow \arg\min_{m \in \mathcal{M}} \mathbb{E}_{S \sim \mathbb{P}_S}[D(\bar{\pi}_0, \pi^m \mid S)]$
3: **for** $t = 1$ **to** $T$ **do**
4:     Observe $S_t$, sample $A_t \sim \pi^m_{\gamma(t)}(\cdot \mid S_t)$, receive $R_t$
5:     **if** $t + 1 = \gamma(t+1)$ **then**
6:         Update model $m \leftarrow \arg\min_{m \in \mathcal{M}} \mathbb{E}_{S \sim \mathbb{P}_S}[D(\bar{\pi}_{\gamma(t+1)}, \pi^m \mid S)]$ *offline*
7:     **end if**
8: **end for**

---

Dropping the time subscript to simplify notation, the imitation learning problem

$$\underset{m \in \mathcal{M}}{\text{minimize}} \, \mathbb{E}_{S \sim \mathbb{P}_S}[D(\bar{\pi}, \pi^m \mid S)]. \tag{3}$$

learns a model $m \in \mathcal{M}$ minimizing a measure of discrepancy $D(\cdot, \cdot \mid S)$ between the two distributions on $\mathcal{A}$, conditional on the context $S$. As the imitation objective equation 3 cannot be computed analytically, we provide efficient approximation algorithms. To instantiate Algorithm 1, we fix Kullback-Leibler (KL) divergence as the notion of discrepancy between probabilities and present finite-sample approximations based on observed contexts and simulated actions from the off-policy Thompson sampling policy $\bar{\pi}_t$. For probabilities $q^1$ and $q^2$ on $\mathcal{A}$ such that $q^1, q^2 \ll \nu$ for some $\sigma$-finite measure $\nu$ on $\mathcal{A}$, the KL divergence between $q^1$ and $q^2$ is $D_{\text{kl}}(q^1 \| q^2) := \int_{\mathcal{A}} \log \frac{dq^1/d\nu}{dq^2/d\nu}(a) d\nu(a)$, where we use $\frac{dq^1}{d\nu}$ and $\frac{dq^2}{d\nu}$ to denote Radon-Nikodym derivatives of $q^1$ and $q^2$ with respect to $\nu$. For two policies $\pi^1$ and $\pi^2$, we define

$$D_{\text{kl}}(\pi^1, \pi^2 \mid S) := D_{\text{kl}}(\pi^1(\cdot \mid S) \| \pi^2(\cdot \mid S)),$$

where we use $\pi^1, \pi^2$ to also denote their conditional densities over $\mathcal{A}$.

The imitation problem equation 3 with $D\left(\cdot, \cdot \mid S\right) = D_{\mathrm{kl}}\left(\cdot, \cdot \mid S\right)$ is equivalent to maximizing log likelihood

$$\underset{m \in \mathcal{M}}{\operatorname{maximize}} \, \mathbb{E}_{S \sim \mathbb{P}_S, \bar{A} \sim \bar{\pi}(\cdot|S)}[\log \pi^m(\bar{A} \mid S)]. \tag{4}$$

In the following, we write $\mathbb{E}[\cdot] = \mathbb{E}_{S \sim \mathbb{P}_S, \bar{A} \sim \bar{\pi}(\cdot|S)}[\cdot]$ for simplicity. In the maximum likelihood estimation (MLE) problem equation 4, the data comprises of context-action pairs. First, contexts are generated under the marginal distribution $S \sim \mathbb{P}_S$ independent of everything else. Conditional on the context, actions are simulated from the batch off-policy Thompson sampling policy $\bar{A} \sim \bar{\pi}(\cdot \mid S)$. The MLE problem equation 4 finds a model $m \in \mathcal{M}$ maximizing the likelihood of observing actions generated by $\bar{\pi}_{\gamma(t)}$.

The imitation objective $m \mapsto \mathbb{E}[\log \pi^m(\bar{A} \mid S)]$ involves an expectation over the unknown marginal distribution of contexts $\mathbb{P}_S$ and actions generated by the Thompson sampling policy $\bar{\pi}(\cdot \mid S)$. Although the expectation over $S \sim \mathbb{P}_S$ involves a potentially high-dimensional integral over an unknown distribution, sampling from this distribution is usually very cheap since the observations $S \sim \mathbb{P}_S$ can be "unsupervised" in the sense that no corresponding action/reward are necessary. For example, it is common for online platforms to maintain a database of features $S$ for all of its users. Using these contexts, we can solve the MLE problem equation 4 efficiently via stochastic gradient descent methods (Kushner & Yin, 2003; Duchi, 2018). In Section 6, we show that it is easy to solve the imitation problem equation 3 to high accuracy by using cheap unsupervised contexts. In Section 5, we show that our imitation algorithm enjoys Bayes regret comparable to that of the batch on-policy Thompson sampling algorithm, up to the sum of single step imitation errors.

For continuous action spaces with a notion of geometry, it is sometimes natural to allow imitation policies to have slightly different support than the Thompson sampling policy. In this scenario, we can instantiate the abstract form of Algorithm 1 with Wasserstein distances as our notion of discrepancy $D\left(\cdot, \cdot \mid s\right)$. The subsequent theoretical development for KL divergences has its analogue for Wasserstein distances, which we outline in Appendix A

## 4  Empirical evaluation

We study the performance of our imitation learning algorithm in terms of cumulative regret / reward and decision-time latency in a number of datasets. Our imitation learning algorithm achieves a significant reduction in latency on all problems and enjoys regret comparable to that of batch on-policy Thompson sampling, avoiding compounding of imitation error over time. Our experiments include a real-world video upload transcoding application for an internet service receiving millions of video upload requests per day.

We include our open-sourced implementation as supplementary materials.

**Datasets**  We compare our imitation algorithm alongside an array of benchmark methods on four problem scenarios. For our first experiment, we study the **wheel bandit problem**, a synthetic problem constructed to require significant exploration (Riquelme et al., 2018). In this two-dimensional problem, there are 5 actions and rarely seen contexts yield high rewards under one context-dependent action. We sample $10,000$ contexts for each trial. Specifically, two-dimensional contexts are sampled in the unit sphere with uniform probability. The first action always has a mean reward of $\mathbb{E}[r(\boldsymbol{s}, a_1)] = 1.2$ independent of the context, and the mean rewards of the other actions depend on the context. If $||\boldsymbol{s}||_2 \le \delta$, then the remaining four actions are non-optimal with a mean reward of 1. If $||\boldsymbol{s}||_2 > \delta$, then one of the remaining actions is optimal—and determined by the sign of the two dimensions of $\boldsymbol{s}$ —with a mean reward of 50. The remaining three actions all have a mean reward of 1. All rewards are observed with zero-mean additive Gaussian noise with standard deviation $\sigma = 0.01$. We set $\delta = 0.95$, which means the probability of sampling a context on the perimeter ($||\boldsymbol{s}||_2 \ge \delta$) where one action yields a large reward is $1 - (0.95)^2 = 0.0975 \approx 10\%$.

For our second problem, we design a contextual bandit problem from a supervised classification task. The **Mushroom UCI Dataset** (mis, 1987) contains 8,124 examples with 22 categorical features about the mushroom and labels indicating if the mushroom is poisonous or not. At each time step, the forager decides whether to eat the mushroom or not and receives a small positive reward for eating a safe mushroom, and a large negative reward for eating an unsafe mushroom. With equal probability, eating a poisonous mushroom lead to illness ($r = -35$) or it may not harm the consumer ($r = 5$), while a nonpoisonous mushroom always

yields a positive reward ($r = 5$). The reward for abstaining is always 0. We sample $50,000$ contexts for each trial.

Next, we turn our attention to a more realistic healthcare scenario, **pharamacological dosage optimization**, where we wish to learn a good dosing policy for Warfarin. Warfarin is one of the most common anticoagulants (blood thinner), often prescribed to patients with atrial fibrillation to prevent strokes (Xiao, 2019). The optimal dosage varies considerably across genetic, demographic, and clinical differences (Bastani & Bayati, 2015). The Warfarin dataset (Xiao, 2019) contains the optimal dosage of Warfarin for $4,788$ patients, which were found via trial and error by physicians. Using a 17-dimensional context vector on patient-specific demographics, medical history, and genetic markers, we construct a contextual bandit benchmark where the action space is a uniformly discretized dosage levels, and rewards are given by absolute deviation from the optimal dosage. We present results for 20 discretized dosage levels, but as we shown in Section E, we observe even bigger latency gains for 50 discretized dosage levels. We present results where we reshuffle contexts for each trial, but again find similar results when $50,000$ contexts are re-sampled each trial.

Finally, we focus on a real-world **video upload transcoding application**, where we study a video upload system for a leading social network platform receiving millions of upload requests on *mobile devices* (see Example 1). The goal is to preserve high quality as much as possible while ensuring upload reliability constraints are met. We use a 38-dimensional context vector representing: (1) Video features: raw bitrate, resolution, file size, duration, codec; (2) Network features: connection type (WiFi/cellular), download bandwidth estimate, upload bandwidth estimate, signal strength; (3) Device features: device model category, operating system, available memory; (4) User features: country, historical upload success rate. There are 7 actions corresponding to a unique (resolution, bitrate) pairs. The actions are ranked ordered in terms of quality: action $i$ yields a video with higher quality than action $j$ if and only if $i \geq j$. If successful, the reward for a successful upload is a positive and monotonically increasing function of the action. The reward for a failed upload is 0.

We evaluate the performance of different contextual bandit algorithms using the unbiased, offline, policy evaluation technique proposed by Li et al. (2011). The method evaluates a contextual bandit algorithm by performing rejection sampling on a stream of logged observation tuples of the form $(S_t, A_t, R_t)$ collected under a uniform random policy. Specifically, the observed tuple is rejected if the logged action does not match the action selected by the algorithm being evaluated. Our dataset contains 8 million observations logged under a uniform random policy. We evaluate each algorithm using the stream of logged data until each algorithm has "observed" $50,000$ *valid* examples.

Our offline evaluation is not meant to suggest offline learning is a valid substitute for online learning algorithms. The cost of randomization and the high level of nonstationarity in the system makes online learning algorithms necessary. We use offline evaluations as an empirically rigorous scientific benchmark that supports and validates our methodological development. Our offline dataset is generated by a particular vertical product, and provided the empirical evidence needed to invest significant resources in implementing the algorithm across multiple products. As the final evaluation, we ran a randomized controlled study as described in the introduction, and observed significant improvements in video quality and topline business metrics (watch time).

**Algorithms and evaluation** For all experiments, we consider models previously found to perform the best in a broad range of benchmark problems, as reported by Riquelme et al. (2018) in their extensive empirical experiments. **Linear-TS** uses an exact Bayesian linear regression to model the reward distribution for each action $a$ independently. This policy evaluates the exact posterior under the assumption that the data for action $a$ were generated from the linear function: $r_a = \boldsymbol{s}^T \boldsymbol{\theta}_a + \varepsilon$ where $\varepsilon \sim \mathcal{N}(0, \sigma_a^2)$. For each action, we independently model the joint distribution, $P(\boldsymbol{\theta}, \sigma^2) = P(\boldsymbol{\theta}|\sigma^2)P(\sigma^2)$ as a normal-inverse-gamma distribution which allows for tractable posterior inference (see Appendix E for closed form expressions). **NeuralLinear-TS** models rewards using a neural network with two 100-unit hidden layers and ReLU activations, but discards the last linear layer and uses the last hidden layer $\boldsymbol{\phi}(\boldsymbol{s})$ as the feature representation for a Linear-TS policy. The neural network takes the context as input, and predicts the reward for each action. The parameters of the neural network are shared for all actions and are learned independently of the Bayesian linear models. **Bootstrap-NN-TS** trains multiple neural networks on bootstrapped observations

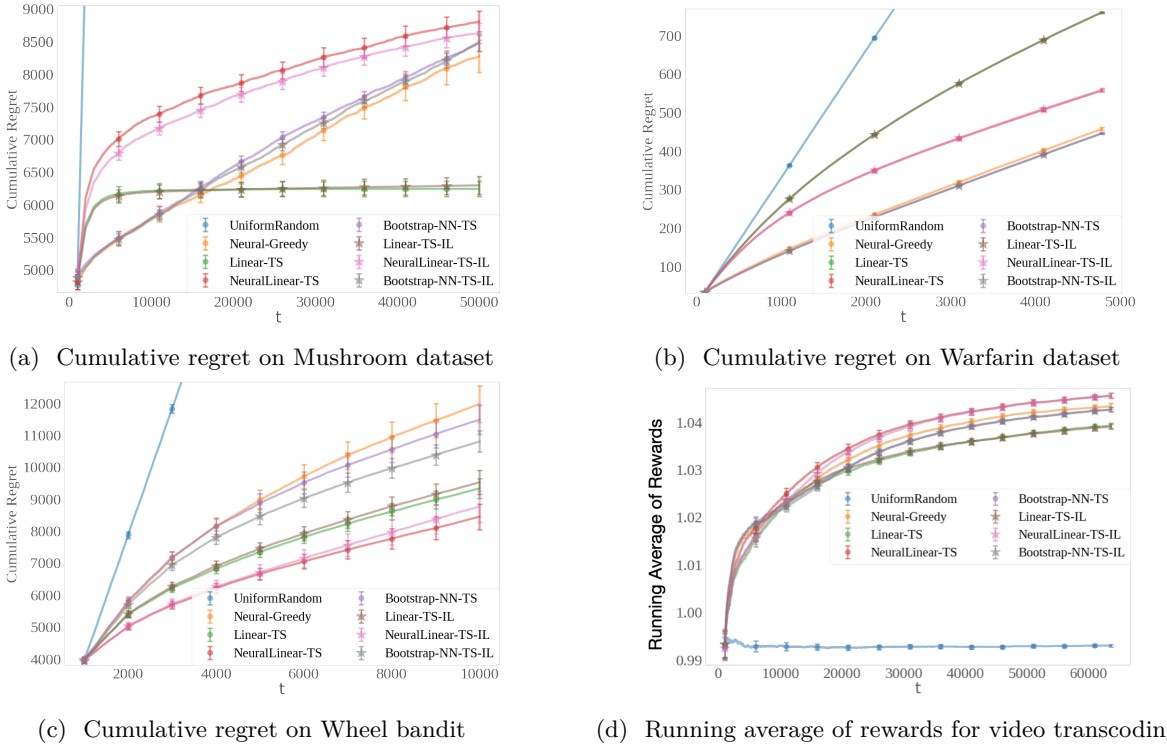

(a)  Cumulative regret on Mushroom dataset

(b)  Cumulative regret on Warfarin dataset

(c)  Cumulative regret on Wheel bandit

(d)  Running average of rewards for video transcoding

Figure 2: We report mean cumulative regret (or running average of rewards for video transcoding), alongside two standard errors over 50 trials (100 trials for the Wheel bandit, due to rarity of large rewards).

and randomly samples a single network to use for each decision. For all of the aforementioned TS policies, **TS-IL** denotes their imitated counterpart. We use a fully-connected neural network to parameterize the policy $\pi^m$ in the imitation learning problem equation 3. The policy representation has two hidden layers with 100 units each, hyperbolic tangent activations on the hidden layers, and a soft-max activation on the output layer to predict a the conditional distribution $P(a|\boldsymbol{s})$ for all $a \in \mathcal{A}$. We compare (batch) Thompson sampling and its imitation counterparts against two additional benchmarks: a random policy (**UniformRandom**) and a greedy policy that uses a feed-froward neural network to model rewards (**Neural-Greedy**).

Policies are updated every 1000 examples (except for the Warfarin problem, where we use update policies every 100 examples due to the small size of the dataset) and are initialized using a uniform random policy before the first batch update. Formally, the mapping $\gamma(t)$ is specified in the definition equation 1, with batch size $B = 1000$ or 100. We detail our hyperparameter choices in Section E: following extensive evaluations by Riquelme et al. (2018), we use their proposed settings for Thompson sampling.

In Figure 2, we show that each TS-IL method achieves performance comparable to its corresponding vanilla TS algorithm on all benchmark problems. We evaluate the cumulative performance at time steps along the entire learning curve, and observe that each TS-IL policy consistently matches its corresponding TS policy over time.

(Approximate) Bayesian inference often requires a substantial amount of compute and memory. We evaluate decision-time latency and time complexity for the specific models being considered, but note that the latency and complexity may be even greater under inference schemes not considered here. We define *decision time latency* as the time required for a policy to select an action when it is queried. While Bootstrap-NN-TS achieves low prediction latency, it requires storing many replicates of the neural network and can significantly increase the memory footprint. On low-end mobile devices, such memory requirements can be prohibitive, limiting the applicability of methods based on bootstrapping; our imitation methods offer a practical and effective alternative.

Table 1: Decision-making latency in milliseconds. All latency measurements were made on a Intel Xeon E5-2680 v4 @ 2.40GHz CPU with 32-bit floating point precision. For each latency measurement, action generation is repeated 100K times and the mean latency and its 2-standard errors are reported.

|  | MUSHROOM | WHEEL | VIDEO TRANSCODE | WARFARIN |
|---|---|---|---|---|
| UNIFORMRANDOM | 0.040 (±0.000) | 0.039 (±0.000) | 0.040 (±0.000) | 0.040 (±0.000) |
| NEURAL-GREEDY | 0.242 (±0.001) | 0.228 (±0.001) | 0.231 (±0.001) | 0.232 (±0.000) |
| LINEAR-TS | 0.715 (±0.001) | 1.142 (±0.001) | 1.575 (±0.002) | 3.963 (±0.002) |
| NEURALLINEAR-TS | 0.826 (±0.001) | 1.492 (±0.001) | 1.931 (±0.002) | 4.814 (±0.004) |
| BOOTSTRAP-NN-TS | 0.235 (±0.001) | 0.235 (±0.001) | 0.236 (±0.001) | 0.226 (±0.001) |
| LINEAR-TS-IL | 0.184 (±0.001) | 0.178 (±0.000) | 0.169 (±0.000) | 0.175 (±0.000) |
| NEURALLINEAR-TS-IL | 0.186 (±0.000) | 0.179 (±0.001) | 0.169 (±0.000) | 0.175 (±0.000) |
| BOOTSTRAP-NN-TS-IL | 0.190 (±0.001) | 0.178 (±0.000) | 0.175 (±0.000) | 0.179 (±0.001) |

Table 1 shows that the imitation policies (TS-IL) have significantly lower decision time latency compared to TS algorithms, often by *over an order of magnitude* on problems with larger action spaces (Warfarin and video upload transcoding). This is because generating an action under the vanilla TS policies requires drawing a sample from the joint posterior $P(\boldsymbol{\theta}_a, \sigma_a^2)$ for each of the actions $a$, which is quadratic with respect to the context dimension for LINEAR-TS or the size of the last hidden layer for NEURALLINEAR-TS. On the other hand, TS-IL simply requires a forward propagation through the policy network and a sample from multinomial sample, both of which are exceedingly cheap. In Section F, we provide a detailed discussion of runtime and memory complexity, including those for alternative model choices.

**Mobile Latency** We report latency on server-grade hardware (Intel Xeon E5-2680) for controlled comparison. On mobile devices, absolute latencies are higher but the relative improvement of our method is preserved or amplified. Based on internal benchmarking on representative mobile devices (iOS and Android), forward passes through our imitation network complete in milliseconds while Thompson sampling operations requiring matrix decompositions can exceed at least an order of magnitude on low-end devices. Memory constraints on mobile devices (typically 2-4GB RAM) further favor our approach: storing a single imitation network versus multiple posterior samples or bootstrap models. These practical constraints motivated our production deployment.

**Randomized Controlled Trial Methodology** Our production RCTs uses Meta's A/B testing platform, which uses a deterministic hashing function based off of a unique user identifier and a unique identifier of the experiment to map subjects to conditions (Bakshy et al., 2014). As such, assignment is independent of any individual characteristics or other treatments. Subjects were randomly assigned to treatment (distilled Thompson sampling) or control (existing rule-based upload policy) groups at the user level, ensuring that each user received consistent treatment throughout the experiment. Experiments ran for 2-4 weeks across more than millions of users per arm, providing sufficient statistical power to detect the reported effect sizes.

## 5 Imitation controls regret

To understand the large practical gains we see in our numerical experiments and randomized controlled study, we now provide some basic theoretical analyses. When the imitation policy generates actions (Algorithm 1), the observations used to update the posterior are different from what the batch Thompson sampling policy would have generated. In this sense, our imitation algorithm does not emulate the batch on-policy Thompson sampling policy, but rather simply mimics its *off-policy* variant where posterior updates are based on the history generated by the imitation policy. In this section, we show how off-policy imitation is sufficient to achieve Bayes regret bounds available for batch *on-policy* Thompson sampling (Russo & Van Roy, 2014), up to the sum of single-step imitation errors. In particular, our results guard against potential exponential compounding of errors that stem from imitating the off-policy variant of batch Thompson sampling.

We show that minimizing the KL divergence equation 3 controls the Bayes regret of the imitation algorithm, justifying the the imitation learning loss equation 3 as a valid objective. First, we relate the performance of our imitation policy with that of the batch off-policy Thompson sampler equation 2 and show batch off-policy Thompson sampling admits a Bayes regret decomposition similar to that for on-policy Thompson sampling (Section 5.1). Building on this observation, we use similar proof techniques for proving Bayes regret bounds on batch on-policy Thompson sampling to provide guarantees for our imitation policy. We substantiate our results in scenarios where batch Thompson sampling is known to provide strong regret bounds (Section 5.2).

## 5.1 Regret decomposition

Since our imitation learning problem equation 3 approximates batch *off-policy* Thompson sampling, a pessimistic view is that any small deviation between the imitation and Thompson sampling policy can exacerbate over time. A suboptimal sequence of actions taken by the imitation policy may deteriorate the performance of the batch off-policy Thompson sampling policy equation 2 updated based on this data, compared to its on-policy counterpart updated based on data collected by itself. Since the imitation policy again mimics this batch off-policy Thompson sampler, this may lead to a negative feedback loop in the worst-case. Our analysis precludes such negative cascades when outcomes are averaged over the prior $P$: the Bayes regret of the imitation policy is comparable to that of the best batch UCB algorithm, up to only the sum of expected discrepancy between the batch off-policy Thompson sampling policy and the imitation learner at each period. In particular, imitation error at each period does not affect the Bayes regret linearly in $T$ as our worst-case intuition suggests, but rather only as a one-time approximation cost. The single-period imitation error can be controlled using cheap unsupervised contexts as we demonstrated in Section 6.

The Bayes regret suffered under the batch off-policy Thompson sampler is a counterfactual quantity as only the imitation policy interacts with the environment. Nevertheless, the fictitious quantity serves an important role in our analysis. Our starting point is that an batch off-policy Thompson sampler enjoys a Bayes regret decomposition similar to *sequential, on-policy* Thompson sampling. Since the off-policy nature of the policy does not affect the Bayes regret decomposition, we are able to bound the Bayes regret of the batch off-policy Thompson sampler using proof techniques developed for *sequential, on-policy* Thompson sampling (Russo & Van Roy, 2014).

Before giving a formal result, we first summarize our approach, which builds on the insights of Russo & Van Roy (2014). We connect the performance of our imitation policy to that of batch off-policy Thompson sampling and in turn relate the latter method's Bayes regret to that of the *best* batch UCB algorithm. Since a similar approach also provides Bayes regret bounds for batch on-policy Thompson sampling, our imitation policy enjoys comparable Bayes regret, up to the sum of single-period imitation errors. Let $U_t(\cdot; H_{\gamma(t)}, S_t) : \mathcal{A} \to \mathbb{R}$ be a sequence of batch upper confidence bounds, constructed using only data collected until the most recent batch $H_{\gamma(t)}$. Let $A_t^{\mathrm{BUCB}}$ be the action taken by the batch UCB policy (BUCB)

$$A_t^{\mathrm{BUCB}} \in \arg\max_{a \in \mathcal{A}} U_t(a; H_{\gamma(t)}, S_t).$$

Recalling the optimal action $A_t^\star \in \arg\max_{a \in \mathcal{A}} f_\theta(a, S_t)$, a typical argument for bounding the regret of a BUCB algorithm proceeds by noting that since $U_t(A_t^{\mathrm{BUCB}}; H_{\gamma(t)}, S_t) \geq U_t(A_t^\star; H_{\gamma(t)}, S_t)$,

$$f_\theta(A_t^\star, S_t) - f_\theta(A_t^{\mathrm{BUCB}}, S_t) \leq f_\theta(A_t^\star, S_t) - U_t(A_t^\star; H_{\gamma(t)}, S_t) + U_t(A_t^{\mathrm{BUCB}}; H_{\gamma(t)}, S_t) - f_\theta(A_t^{\mathrm{BUCB}}, S_t).$$

Taking expectations and summing over $t = 1, \ldots, T$, BayesRegret $\left(T, \{\pi_t^{\mathrm{BUCB}}\}_{t \in \mathbb{N}}\right)$ is bounded by

$$\sum_{t=1}^T \mathbb{E}[f_\theta(A_t^\star, S_t) - U_t(A_t^\star; H_{\gamma(t)}, S_t)] + \sum_{t=1}^T \mathbb{E}[U_t(A_t^{\mathrm{BUCB}}; H_{\gamma(t)}, S_t) - f_\theta(A_t^{\mathrm{BUCB}}, S_t)].$$

If the upper confidence bound property holds uniformly over the actions so that $U_t(a; H_{\gamma(t)}, S_t) \geq f_\theta(a, S_t)$ for all $a \in \mathcal{A}$ with high probability, the first term in the above regret decomposition can be seen to be nonpositive. To bound the second term, a canonical proof notes each upper confidence bound is not too far away from the population mean $f_\theta(A_t^{\mathrm{BUCB}}, S_t)$. Russo & Van Roy (2014)'s key insight was that (sequential on-policy)

Thompson sampling admits an analagous Bayes regret decomposition as above, but with respect to *any* UCB sequence. This allows leveraging arguments that bound the (frequentist) regret of a UCB algorithm to bound the Bayes regret of Thompson sampling. Since the Bayes regret decomposition for Thompson sampling holds for *any* UCB sequence, the performance of Thompson sampling enjoys Bayes regret guarantees of the best UCB algorithm.

By connecting the performance of our imitation policy to that of *batch off-policy Thompson sampling*, we show that a similar Bayes regret decomposition can be leveraged despite its off-policy nature. Recall that we denote $\bar{A}_t \sim \bar{\pi}_{\gamma(t)}(\cdot \mid S_t)$, the action generated by the batch off-policy Thompson sampler. See Section B for the proof of the following result.

**Lemma 1.** *Let $\{\pi_{\gamma(t)}\}_{t \in \mathbb{N}}$ and $U_t(\cdot; H_{\gamma(t)}, S_t)$ be any sequence of batch policies and UCBs (adapted to the history $H_{\gamma(t)}$). If $\mathbb{E}[\sup_{a \in \mathcal{A}} f_\theta(a, S)^2] =: L^2 < \infty$,*

$$\text{BayesRegret}\left(T, \{\pi_{\gamma(t)}\}_{t \in \mathbb{N}}\right) \leq \underbrace{\sum_{t=1}^{T} \mathbb{E}[f_\theta(A_t^\star, S_t) - U_t(A_t^\star; H_{\gamma(t)}, S_t)] + \sum_{t=1}^{T} \mathbb{E}[U_t(\bar{A}_t; H_{\gamma(t)}, S_t) - f_\theta(\bar{A}_t, S_t)]}_{\textit{(a): regret decomposition for any batch UCB algorithm}}$$

$$+ \underbrace{L \sum_{t=1}^{T} \sqrt{\frac{1}{2} \mathbb{E}\left[D_{\text{kl}}\left(\bar{\pi}_{\gamma(t)}, \pi_{\gamma(t)} \mid S_t\right)\right]}}_{\textit{(b): imitation error}}. \tag{5}$$

The Bayes regret decomposition equation 5 shows that performance analysis of any batch UCB algorithm can characterize the regret of our imitation policy. In this sense, the imitation policy achieves regret comparable to the *optimal* batch UCB algorithm, up to the sum of single-period imitation errors. As we detail shortly in a general modeling scenario based on contextual Gaussian processes, term (a) can be bounded using canonical batch UCB proofs. Term (b) can be controlled by our imitation learning algorithm (Algorithm 1) and its empirical approximation as seen in Section 6. Although this term scales as $O(T/\sqrt{N})$, we argue that the seemingly linear dependence on $T$ is not of material concern. In large-scale internet applications, the number of unsupervised contexts $N$ is very large as they can simply be read off of a database of user information ($N \approx 10 - 100M$). The number of policy updates $T$ is often orders of magnitude smaller (hundreds) in a typical product lifecycle due to operational challenges in deploying a policy. Thus, the term (b) can be made relatively small using big datasets and powerful overparameterized imitation models using the results in Section 6. The fact that we are studying Bayes regret, as opposed to the frequentist regret, plays an important role in the above decomposition. We conjecture that in the worst-case, imitation error at any period (and consequently suboptimal exploration) can each linearly compound over time, leading to a prohibitive quadratic dependence on $T$. It remains open whether specific problem structures can provably preclude such negative feedback loops uniformly over $\theta$.

The main insight here is that the Bayes regret analysis averages over the parameter drawn from the prior and under this averaging, the quality of the posterior—whether it was updated using on-policy or off-policy data—does not directly affect the regret decomposition. What matters is the current discrepancy between the imitation policy and the Thompson sampling policy at each time step, not the cumulative history of discrepancies. This is because Thompson sampling's optimality property (selecting actions according to their posterior probability of being optimal) holds regardless of how the posterior was formed. As a result, while off-policy data may lead to a posterior that is less informative than the on-policy counterpart, it does not systematically bias the Thompson sampling policy in a way that compounds errors. Each period's imitation error contributes additively to the Bayes regret, rather than multiplicatively.

## 5.2 Regret bounds for contextual Gaussian processes

We now show concrete performance guarantees for our imitation algorithm by using instance-independent (gap-independent) Bayes regret bounds for batch off-policy Thompson sampling. Despite its counterfactual nature, the decomposition equation 5 enables us to control it using identical proof techniques for controlling

the Bayes regret of batch *on-policy* Thompson sampling. This program allows us control over the term $(a)$ in the decomposition equation 5.

We consider a general setting where the mean reward function $(a, s) \mapsto f_\theta(a, s)$ can be modeled as a sample path of a Gaussian process, with potentially continuous action and context spaces. Formally, we assume that $(a, s) \mapsto f_\theta(a, s)$ is sampled from a Gaussian process on $\mathcal{A} \times \mathcal{S}$ with mean function $\mu(a, s)$ and covariance function (kernel)

$$\Sigma((a, s), (a', s')) := \mathbb{E}[(f_\theta(a, s) - \mu(a, s))(f_\theta(a', s') - \mu(a', s'))].$$

We assume that the decision maker observes rewards

$$R_t = f_\theta(A_t, S_t) + \epsilon_t,$$

where the noise $\epsilon_t \overset{\text{iid}}{\sim} N(0, \sigma^2)$ are independent of everything else. Given these rewards, we are interested in optimizing the function $a \mapsto f_\theta(a, S_t)$ for each observed context $S_t$ at time $t$.

Modeling mean rewards as a Gaussian process is advantageous since we can utilize analytic formulae to update the posterior at each step. Since $f_\theta(a, s)$ follows a Gaussian process, its posterior is also a Gaussian process with mean and variance is given by

$$\mu_t(a, s) := \mathbb{E}[f_\theta(a, s) \mid H_t] = \Sigma_t(a, s)^\top (K_t + \sigma^2 I)^{-1} \vec{R}_t,$$
$$\sigma_t^2(a, s) := \text{Var}(f_\theta(a, s) \mid H_t) = \Sigma((a, s), (a, s)) - \Sigma_t(a, s)^\top (K_t + \sigma^2 I)^{-1} \Sigma_t(a, s)$$

where $\Sigma_t(a, s) := [\Sigma((A_j, S_j), (a, s))]_{1 \le j \le t-1}$, $K_t := [k((A_i, S_i), (A_j, S_j))]_{1 \le i,j \le t-1}$ and $\vec{R}_t = [R_j]_{1 \le j \le t-1}$. For large-scale applications, we can parameterize our kernels by a neural network and leverage the recently developed interpolations techniques to perform offline posterior updates (Wilson & Nickisch, 2015; Wilson et al., 2015; 2016).

We leverage regret bound techniques for batch UCB algorithms (Desautels et al., 2014) to bound the term $(a)$ in the Bayes regret decomposition equation 5. This term is controlled by the maximal amount of information on the optimal action that can be gained after $T$ time steps. Recall the definition of (conditional) mutual information between two random vectors

$$I(Z, Y) := D_{\text{kl}}(P_{Z,Y} \| P_Z \times P_Y) \quad \text{and} \quad I(Z, Y \mid W) := D_{\text{kl}}(P_{Z,Y|W} \| P_{Z|W} \times P_{Y|W})$$

We define the maximal possible information gain after $T$ time steps as

$$\gamma_T := \sup_{\mathcal{X} \subseteq \mathcal{A} \times \mathcal{S} : |\mathcal{X}| = T} I(\vec{R}_\mathcal{X}, f_\mathcal{X})$$

where $\vec{R}_\mathcal{X} = \{f_\theta(x) + \epsilon_x\}_{x \in \mathcal{X}}$ and $f_\mathcal{X} = \{f_\theta(x)\}_{x \in \mathcal{X}}$. For popular Gaussian and Matern kernels, Srinivas et al. (2012) has shown that the maximal information gain can be bounded explicitly; we summarize these bounds shortly.

Due to the batched nature of Algorithm 1, we further need to control the maximal information gain in a single batch, assuming that the (time-varying) batch size is uniformly bounded by some constant $B$.

**Assumption A.** *Let $\gamma(t+1) - \gamma(t) \le B$ for $1 \le t \le T$ and let $\eta_B$ be a constant satisfying*

$$\max_{\mathcal{X} \subseteq \mathcal{A} \times \mathcal{S} : |\mathcal{X}| \le B} I\left(\vec{R}_\mathcal{X}, f_\mathcal{X} \mid \vec{R}_{\gamma(t)}\right) \le \frac{1}{2} \log(\eta_B) \quad \text{for all } 1 \le t \le T \tag{6}$$

*where $\vec{R}_\mathcal{X} = \{f_\theta(x) + \epsilon_x\}_{x \in \mathcal{X}}$, $\vec{R}_{\gamma(t)} = \{R_1, \ldots, R_{\gamma(t)-1}\}$, and $f_\mathcal{X} = \{f_\theta(x)\}_{x \in \mathcal{X}}$.*

For a compact action space $\mathcal{A} \subset \mathbb{R}^d$, term $(a)$ in the decomposition equation 5 is bounded by $O\left(\sqrt{d \eta_B \gamma_t T (\log T)^d}\right)$. Our proof relies on the batch upper confidence bound

$$U_t(a; H_{\gamma(t)}, s) := \mu_{\gamma(t)}(a, s) + \sqrt{\beta_t} \sigma_{\gamma(t)}(a, s) \quad \text{where} \quad \beta_t = 2 \log((T^4 r d)^d T^2) \tag{7}$$

We use $L_f$ to denote the (random) Lipschitz constant of the map $a \mapsto f_\theta(a, s)$

$$L_f := \sup_{s \in \mathcal{S}} \sup_{a, a' \in \mathcal{A}} \frac{|f_\theta(a, s) - f_\theta(a', s)|}{\|a - a'\|_1}.$$

Standard arguments from Gaussian process theory show $\mathbb{E}[L_f^2] < \infty$ holds whenever $\mu(\cdot)$ and $\Sigma(\cdot, \cdot)$ are 4 times continuously differentiable (Ghosal et al., 2006, Theorem 5).

**Theorem 1.** *For $\mathcal{A} \subseteq [0, r]^d$ for some $r > 0$, let Assumption A hold. Assume that*

$$c_1 := \sup_{a \in \mathcal{A}, s \in \mathcal{S}} |\mu(a, s)| < \infty, \qquad c_2 := \sup_{a, a' \in \mathcal{A}, s, s' \in \mathcal{S}} \Sigma((a, s), (a', s')) < \infty,$$

*and let $L^2 := \mathbb{E}\left[\sup_{a \in \mathcal{A}, s \in \mathcal{S}} f_\theta(a, s)^2\right]$ as before. If $\mathbb{E}[L_f^2] < \infty$, there is a universal constant $C > 1$ such that*

$$\text{BayesRegret}\,(T, \pi) \leq C\mathbb{E}[L_f] + Cc_2 + Cd\log(rd)\left(c_1\sqrt{\mathbb{E}[L_f]} + c_3\sqrt{\mathbb{E}[L_f^2]}\right)$$

$$+ \left(T\eta_B\gamma_T \frac{d\log T + d\log rd}{\log(1 + \sigma^{-2})}\right)^{1/2} + (L + \sqrt{c_2\beta_T}) \sum_{t=1}^{T} \sqrt{\frac{1}{2}\mathbb{E}\left[D_{\mathrm{kl}}\left(\bar{\pi}_{\gamma(t)}, \pi_{\gamma(t)} \mid S_t\right)\right]}.$$

See Section C.1 for the proof.

**Bounds on $\gamma_T$**   To obtain concrete bounds on the maximal information gain $\gamma_T$, we focus on the popular and flexible linear, Gaussian and Matern kernels

$$\Sigma_l(x, x') := x^\top x', \Sigma_g(x, x') := \exp\left(-\frac{\|x - x'\|^2}{2l^2}\right),$$

$$\Sigma_m(x, x') := \frac{2^{1-\nu}}{\Gamma(\nu)} r^\nu B_\nu(r) \quad \text{where } r = \frac{\sqrt{2\nu}}{l}\|x - x'\|,$$

where we used $B(\cdot)$ and $\Gamma(\cdot)$ to denote the Besel and Gamma functions respectively. To ease notation, we let $\kappa$ denote the dimension of the underlying space, and define

$$\mathfrak{M}(\Sigma_l, T) := \kappa\log T, \quad \mathfrak{M}(\Sigma_g, T) := (\log T)^{\kappa+1}, \quad \mathfrak{M}(\Sigma_m, T) := T^{\frac{\kappa^2+\kappa}{\kappa^2+\kappa+2\nu}}\log T.$$

We have the following bound on $\gamma_T$ for linear, Gaussian, and Matern kernels; the bound is a direct consequence of Krause & Ong (2011, Theorem 2) and Srinivas et al. (2012, Theorem 5).

**Lemma 2.** *Let $\mathcal{A} \subseteq \mathbb{R}^d$ and $\mathcal{S} \subseteq \mathbb{R}^{d'}$ be convex and compact. Let the kernel $\Sigma$ be given by $\Sigma((a, s), (a', s')) := \Sigma_A(a, a') + \Sigma_S(s, s')$. Then, $\gamma_T = O\left(\mathfrak{M}(\Sigma_A, T) + \mathfrak{M}(\Sigma_S, T) + \log T\right)$.*

**Bounds on $\eta_B$**   To control the Bayes regret of batch off-policy Thompson sampling, it remains to control the per batch information gain $\eta_B$. Our development so far allows us to use techniques developed for on-policy Thompson sampling to bound this quantity. A naive bound for the per batch information gain $\eta_B$ is $\eta_B \leq \exp(2\gamma_B)$, which can be prohibitively large in large batch scenarios. Towards tighter theoretical control, we use a clever initialization scheme due to Desautels et al. (2014). While we conjecture that batch Thompson sampling will perform well even without such a careful initialization scheme, we are unable to theoretically confirm the conjecture and leave it as future work.

We initialize our algorithm by targeting $T_{\text{init}}$ users/contexts who suffer the highest uncertainty in their reward. Considering the initialization index set $t \in \{-T_{\text{init}} + 1, \ldots, 0\}$, the posterior variance does not depend on previous rewards

$$\sigma_t^2(a, s) := \text{Var}(f_\theta(a, s) \mid H_t) = \Sigma((a, s), (a, s)) - \Sigma_t(a, s)^\top (K_t + \sigma^2 I)^{-1}\Sigma_t(a, s),$$

where $\Sigma_{(a,s)} := [\Sigma((A_j, S_j), (a, s))]_{-T_{\text{init}}+1 \leq j \leq t-1}$ and $K_t := [k((A_t, S_t), (A_j, S_j))]_{-T_{\text{init}}+1 \leq j \leq t-1}$. Thus, before engaging with the environment we can sequentially calculate

$$(A_t^{\text{init}}, S_t^{\text{init}}) \in \underset{a \in \mathcal{A}, s \in \mathcal{S}}{\arg\max} \, \sigma_t^2(a, s) \qquad \text{for} \quad t = -T_{\text{init}} + 1, \ldots, 0$$

We initially target users/contexts $S_t^{\text{init}}$ in the database with actions $A_t^{\text{init}}$ for $t = -T_{\text{init}} + 1, \ldots, 0$. Using the history $H_t = \{S_i, A_i, R_i\}_{i=-T_{\text{init}}+1}^{t-1}$, we redefine Thompson sampling and Algorithm 1 with initialization data. The following result shows that this initialization procedure controls the per batch information gain $\eta_B$. For simplicity, we consider combinations of linear or Gaussian kernels and define $\bar{d} := \max(d, d')$. Recalling the batch upper confidence bound equation 7, the following result is a direct consequence of Desautels et al. (2014, Lemma 4, Theorem 5); an analogous bound holds for Matern kernels, but we omit it for brevity.

**Proposition 2.** *Let the conditions of Theorem 1 hold and let $\Sigma_A, \Sigma_S \in \{\Sigma_l, \Sigma_g\}$. Consider the initialization procedure described in the previous paragraph with $T_{\text{init}}$ periods. There is a constant $C > 0$ such that if we set $T_{\text{init}} = C^{\bar{d}+1} B (\log B)^{\bar{d}+1}$, then*

$$\text{BayesRegret}\left(T, \{\pi_{\gamma(t)}\}_{t \in \mathbb{N}}\right) = \sum_{t=1}^{T} \mathbb{E}_{\theta \sim P} \left[ \max_{a \in \mathcal{A}} f_\theta(a, S_t) - f_\theta(A_t, S_t) \right]$$

$$\leq O\left( \exp(\bar{d}^{\bar{d}}) \sqrt{\bar{d} T (\log T)^{\bar{d}+1}} \right) + (L + \sqrt{c_2 \beta_T}) \sum_{t=1}^{T} \sqrt{\frac{1}{2} \mathbb{E}\left[ D_{\text{kl}}\left( \bar{\pi}_{\gamma(t)}, \pi_{\gamma(t)} \mid S_t \right) \right]}.$$

The first term bounds the Bayes regret of batch on-policy Thompson sampling; in comparison, sequential on-policy Thompson sampling (Krause & Ong, 2011) achieves Bayes regret $O\left( \sqrt{\bar{d} T (\log T)^{\bar{d}+1}} \right)$.

**Discussion of Assumptions**   Our theoretical results rely on several assumptions that merit discussion on their applicability. The Gaussian process assumption is standard in the Bayesian optimization literature and is reasonable when rewards are smooth functions of context-action pairs. In practice, the GP assumption serves more as a theoretical vehicle; our empirical results (Section 4) demonstrate strong performance even when using linear and neural network-based models that may not strictly satisfy GP assumptions. More specifically, the smoothness conditions on kernels ensure bounded Lipschitz constants, which popular kernels including Gaussian (RBF), Matérn with $\nu > 2$, and polynomial kernels satisfy.

The theoretical initialization scheme targeting high-uncertainty contexts is primarily a proof technique ensuring bounded per-batch information gain. In practice, uniform random initialization performs well (as shown in our experiments) because real-world contexts naturally provide sufficient coverage. The gap between our theoretical initialization and practical random initialization remains an interesting open question.

Our results require barring two types of misspecification: first, the imitation model class must contain the true Thompson sampling policy; second, the Thompson sampling policy's prior must be well-specified. While the former is relatively mild given how well modern neural networks with sufficient capacity can approximate complex distributions, the second inherits all the non-robustness issues of Thompson sampling. More generally, our imitation approach will inherit any limitation of the original model class.

## 6   Generalization guarantees for imitation learning

In this section, we show that solving an empirical approximation of the imitation problem equation 4 can control the imitation objective. From results in the previous section, this in turn shows that the regret can be controlled when we have many contexts. Given i.i.d. observations of (potentially unsupervised contexts) $S_i \overset{\text{iid}}{\sim} P_S$, we solve the empirical approximation to the imitation problem equation 4

$$\widehat{m} \in \underset{m \in \mathcal{M}}{\arg\max} \, \frac{1}{N} \sum_{i=1}^{N} \frac{1}{N_a} \sum_{j=1}^{N_a} \log \pi^m(\bar{A}_{ij} \mid S_i), \tag{8}$$

where we simulate actions from the batch off-policy Thompson sampling equation 2

$$\bar{A}_{ij} \sim \bar{\pi}_{\gamma(t)}(\cdot \mid S_i) \quad j = 1, \ldots, N_a$$

for each context $S_i$. Since actions can be simulated *offline* in a parallel manner, we can efficiently generate a large number of actions $N_a$.

In what follows, we assume that our imitation model class is *well-specified*, so that there exists $m^\star \in \mathcal{M}$ satisfying $\bar{\pi} = \pi^{m^\star}$, where we omitted the subscript and denote $\bar{\pi} = \bar{\pi}_{\gamma(t)}$ to ease notation. This is often a reasonable assumption as we consider expressive model classes such as nonparametric models involving reproducing kernel Hilbert spaces. With a well-specified imitation model, we prove with probability at least $1 - \delta$,

$$\mathbb{E}_{S \sim \mathbb{P}_s} D_{\mathrm{kl}}\left(\bar{\pi}, \pi^{\widehat{m}} \mid S\right) \lesssim \frac{1}{N}\left(\mathfrak{Comp}_N + \log\frac{1}{\delta}\right) + \frac{\mathfrak{Comp}_{N,N_a}}{\sqrt{NN_a}}, \tag{9}$$

for some complexity measures $\mathfrak{Comp}_N$ and $\mathfrak{Comp}_{N,N_a}$ associated with the imitation model class $\mathcal{M}$. Here, the notation $\lesssim$ denotes inequality up to a universal constant. In typical internet applications, the number of unsupervised contexts $N$ is exceedingly large, and the imitation error equation 9 can be made vanishingly small.

The key challenges to showing the preceding result are twofold: 1) the empirical procedure equation 8 employs non-i.i.d. samples $(S_i, \bar{A}_{ij})$, so standard concentration results do not apply, and 2) the bound equation 9 scales with the "fast rate" $1/N$, rather than the canonical parametric rate $1/\sqrt{N}$. To overcome the first challenge, our proof carefully derives concentration inequalities for the two-step sampling process where nature generates $S_i \overset{\mathrm{iid}}{\sim} \mathbb{P}_S$, and for each $S_i$ we simulate $A_{ij} \overset{\mathrm{iid}}{\sim} \bar{\pi}(\cdot \mid S_i)$ via posterior sampling. To prove the fast rate of convergence $1/N$, we use an elaborate localization-based proof approach (Bartlett et al., 2005) which exploits the fact that the complexity of the function class $(s, a) \mapsto \log \pi^m(a \mid s)$ may be substantially smaller on a neighborhood of the optimum $m^\star$, compared to over the entire model space $m \in \mathcal{M}$.

To formalize our arguments, recall the standard notion of Rademacher complexity: for a fixed $\xi_1, \ldots, \xi_n$ and i.i.d. random signs $\varepsilon_i \in \{-1, 1\}$ (Rademacher variables) that are independent of the $\xi_i$'s, the empirical Rademacher complexity of the class of functions $\mathcal{G} \subseteq \{g : \Xi \to \mathbb{R}\}$ is

$$\mathfrak{R}_n(\mathcal{G}) := \mathbb{E}_\varepsilon\left[\sup_{g \in \mathcal{G}} \frac{1}{n}\sum_{i=1}^n \varepsilon_i g(\xi_i)\right].$$

A function $\psi : \mathbb{R}_+ \to \mathbb{R}_+$ is *sub-root* (Bartlett et al., 2005) if it is nonnegative, nondecreasing, and $r \mapsto \psi(r)/\sqrt{r}$ is nonincreasing for all $r > 0$. This analytic notion guarantees that any non-constant sub-root function $\psi$ is continuous, and has a unique positive fixed point $r^\star = \psi(r^\star)$, where $r \geq \psi(r)$ for all $r \geq r^\star$. Let $\psi_n : \mathbb{R}_+ \to \mathbb{R}_+$ be a sub-root upper bound on the localized Rademacher complexity

$$\psi_n(r) \geq \mathbb{E}[\mathfrak{R}_n(\{g \in \mathcal{G} : \mathbb{E}[g^2] \leq r\})]. \tag{10}$$

(The localized Rademacher complexity itself is sub-root.) Fixed points of $\psi_n$ characterize uniform concentration guarantees; see Bartlett et al. (2005) and Koltchinskii (2006) for a detailed analysis of localized Rademacher complexities.

The Rademacher complexity of the following set of functions controls the generalization performance of the empirical imitation model equation 8

$$\mathcal{G}_1 := \left\{s \mapsto \mathbb{E}_{\bar{A} \sim \bar{\pi}(\cdot \mid s)}\left[\log\frac{\bar{\pi}(\bar{A} \mid s)}{\pi^m(\bar{A} \mid s)}\right] : m \in \mathcal{M}\right\}$$

$$\mathcal{G}_2(s) := \{a \mapsto \log \pi^m(a \mid s) : m \in \mathcal{M}\}$$

$$\mathcal{G}_3 := \{(a, s) \mapsto \log \pi^m(a \mid s) : m \in \mathcal{M}\}.$$

We let $r_N^\star$ be the unique fixed point of the sub-root function $\psi_n$ satisfying the bound equation 10 for $\mathcal{G} = \mathcal{G}_1$

$$r_N^\star = \psi_n(r_N^\star) \quad \text{where} \quad \psi_n(r) \geq \mathbb{E}[\mathfrak{R}_n(\{g \in \mathcal{G}_1 : \mathbb{E}[g^2] \leq r\})]$$

For any fixed context $s \in \mathcal{S}$, using i.i.d. random signs $\varepsilon_j$, we write

$$\mathfrak{R}_N \mathcal{G}_2(s) := \mathbb{E}_\epsilon \left[ \sup_{m \in \mathcal{M}} \frac{1}{N_a} \sum_{j=1}^{N_a} \varepsilon_j \log \pi^m(\bar{A}_j \mid s) \right].$$

For $\mathcal{G}_3$, using i.i.d. random signs $\varepsilon_{ij}$ we still write

$$\mathfrak{R}_{NN_a} \mathcal{G}_3 := \mathbb{E}_\epsilon \left[ \sup_{m \in \mathcal{M}} \frac{1}{N} \sum_{i=1}^{N} \frac{1}{N_a} \sum_{j=1}^{N_a} \varepsilon_{ij} \log \pi^m(\bar{A}_{ij} \mid S_i) \right].$$

Our main result in this section shows that the imitation error of the empirical solution equation 8 is $O_p \left( N^{-1} + N^{-1/2} N_a^{-1/2} \right)$. See Section D.1 for the proof.

**Theorem 3.** *Let there exist a $m^\star \in \mathcal{M}$ such that $\bar{\pi} = \pi^{m^\star}$. Assume $|\log \pi^m(a \mid s)| \leq M$ for all $a \in \mathcal{A}, s \in \mathcal{S}, m \in \mathcal{M}$. There is a numerical constant $C > 0$ s.t. with probability at least $1 - 2e^{-z}$*

$$\mathbb{E}\left[ D_{\mathrm{kl}} \left( \bar{\pi}, \pi^{\widehat{m}} \mid S \right) \right] \leq C \left( \frac{1}{M} r_N^\star + \frac{Mt}{N} + \sqrt{\frac{z}{N}} \sup_{s \in \mathcal{S}} \mathbb{E}_{\bar{A}_j \overset{\mathrm{iid}}{\sim} \bar{\pi}(\cdot \mid s)} [\mathfrak{R}_{N_a}(\mathcal{G}_2(s))] + \mathbb{E}[\mathfrak{R}_{NN_a}(\mathcal{G}_3)] \right).$$

For finite-dimensional model classes with bounded VC-dimension, standard arguments bound the Rademacher complexity terms in the above theorem (van der Vaart & Wellner, 1996, Ch 2.6). Denoting by $\mathsf{VC}(\cdot)$ the VC-dimension, we have

$$\sup_{s \in \mathcal{S}} \mathbb{E}_{\bar{A}_j \overset{\mathrm{iid}}{\sim} \bar{\pi}(\cdot \mid s)} [\mathfrak{R}_{N_a}(\mathcal{G}_2(s))] \leq M \sqrt{\frac{\sup_{s \in \mathcal{S}} \mathsf{VC}(\mathcal{G}_2(s))}{N_a}} \quad \text{and} \quad \mathbb{E}[\mathfrak{R}_{NN_a}(\mathcal{G}_3)] \leq M \sqrt{\frac{\mathsf{VC}(\mathcal{G}_3)}{NN_a}}.$$

Moreover, Corollary 3.7 of Bartlett et al. (2005) implies that $r_N^\star \asymp \frac{M \mathsf{VC}(\mathcal{G}_1) \log(N/\mathsf{VC}(\mathcal{G}_1))}{N}$. Plugging these bounds in Theorem 3, we obtain the previously claimed convergence rate equation 9.

Due to the generality of our localized Rademacher complexity approach, we can provide imitation guarantees for substantially larger and more expressive *nonparametric* model classes. We consider a reproducing kernel Hilbert space (RKHS) $\mathcal{H}$ defined over a kernel $k : \Xi \times \Xi \to \mathbb{R}_+$ (Berlinet & Thomas-Agnan, 2004). For such nonparametric models, standard covering number bounds are loose (Kühn, 2011), while localized arguments can still provide fast concentration (Mendelson, 2003). Consider a RKHS with norm $\|\cdot\|_{\mathcal{H}}$ and evaluation kernel $k(\cdot, \cdot)$. Mercer's theorem (Cristianini & Shawe-Taylor, 2004) states that the integral operator $T_k : L^2(\Xi, P) \to L^2(\Xi, P), T_k(h)(\xi) = \int h(\xi') K(\xi, \xi') dP(\xi')$ is compact, and we have the eigenbasis expansion $k(\xi, \xi') = \sum_{j=1}^{\infty} \lambda_j \phi_j(\xi) \phi_j(\xi')$ where $\lambda_j$ are eigenvalues of $T$ sorted in decreasing order and $\phi_j$ give an orthonormal decomposition in $L^2(\mathcal{Z}, P)$.

Let $k_{\mathcal{S}} : \mathcal{S} \times \mathcal{S} \to \mathbb{R}_+$ and $k_{\mathcal{A}} : \mathcal{A} \times \mathcal{A} \to \mathbb{R}_+$ be kernels on $\mathcal{S}$ and $\mathcal{A}$ respectively, and let us denote by $\mathbb{B}_{\mathcal{S}}$ and $\mathbb{B}_{\mathcal{A}}$ the unit ball in the respective RKHS's. The kernels $k_{\mathcal{S}}$ and $k_{\mathcal{A}}$ induce a RKHS over functions on $\mathcal{S} \times \mathcal{A}$ formed with the kernel $k((s, a), (s', a')) = k_{\mathcal{S}}(s, s') + k_{\mathcal{A}}(a, a')$; we denote the unit ball in this space by $\mathbb{B}_{\mathcal{S} \times \mathcal{A}}$. For simplicity, we assume that the function classes $\mathcal{G}_1, \mathcal{G}_2(s)$, and $\mathcal{G}_3$ belong in a unit ball in appropriately defined RKHS's

$$\mathcal{G}_1 \subset \mathbb{B}_{\mathcal{S}}, \quad \mathcal{G}_2(s) \subset \mathbb{B}_{\mathcal{A}} \text{ for all } s \in \mathcal{S}, \quad \mathcal{G}_3 \subset \mathbb{B}_{\mathcal{S} \times \mathcal{A}}.$$

For RKHS-based models, the rate of decay of the eigenvalues of $T_{k_{\mathcal{S}}}$ controls the rate of convergence in Theorem 3. For example, eigenvalues of the popular Gaussian kernel $k(\xi, \xi') = \exp(-\frac{1}{2} \|\xi - \xi'\|_2^2)$ decay exponentially fast $\lambda_j \lesssim e^{-j^2}$ (Mendelson, 2003). Eigenvalues of kernel operators $T_k$ for Sobolev spaces (Birman & Solomjak, 1967; Gu, 2002) decay polynomially fast $\lambda_j \lesssim j^{-2\beta}$, where $\beta > \frac{1}{2}$ is the smoothness level. e.g., in 1-dimension, the first-order Sobolev kernel $k(\xi, \xi') = 1 + \min\{\xi, \xi'\}$ where $\beta = 1$ generates RKHS of Lipschitz functions. We prove the below corollary in Section D.2.

**Corollary 1.** *Assume* $\sup_{s\in\mathcal{S}} k_{\mathcal{S}}(s,s) + \sup_{a\in\mathcal{A}} k_{\mathcal{A}}(a,a) \leq B$ *for some* $B > 0$. *If the eigenvalues of* $T_{k_{\mathcal{S}}}$ *decay as* $\lambda_j \lesssim e^{-j^2}$, *there is a numerical constant* $C > 0$ *s.t. with probability at least* $1 - 2e^{-z}$

$$\mathbb{E}\left[D_{\mathrm{kl}}\left(\bar{\pi}, \pi^{\widehat{m}} \mid S\right)\right] \leq C \frac{Mz + \sqrt{\log N}}{N} + CMB\sqrt{\frac{z+1}{NN_a}}.$$

*If the eigenvalues of* $T_{k_{\mathcal{S}}}$ *decay as* $\lambda_j \lesssim j^{-2\beta}$ *for some* $\beta > 1/2$, *then there is another numerical constant* $C > 0$ *such that with probability at least* $1 - 2e^{-z}$

$$\mathbb{E}\left[D_{\mathrm{kl}}\left(\bar{\pi}, \pi^{\widehat{m}} \mid S\right)\right] \leq C\left(\frac{Mz}{N} + N^{\frac{-2\beta}{2\beta+1}} + MB\sqrt{\frac{z+1}{NN_a}}\right).$$

## 7 Discussion

In this paper, we used imitation learning to operationalize Thompson sampling, allowing it to scale to applications where latency and software complexity are of concern. We demonstrated that imitation learning provides a simple, practical, and efficient method with desirable regret properties. By distilling the Thompson sampling policy into easy-to-deploy explicit policy representations (e.g. neural networks), we allow state-of-the-art Bayesian approaches to be used in contextual bandit problems. We hope that this work facilitates applications of modern deep learning-based Bayesian approaches to large-scale contextual bandit problems.

Our algorithmic approach shifts computation from online latency-critical to offline batch processing environments, and is not meant to be taken as a universally dominating intervention. This trade-off is most useful when offline computation is embarrassingly parallel and can leverage powerful backend servers, whereas online computation must execute on resource-constrained mobile devices with strict latency requirements. For example, in our video transcoding deployment, $N$ is chosen to be sufficiently large for accurate imitation (typically $N = 10^5$ contexts) while remaining computationally feasible for offline batch processing. With modern distributed computing infrastructure, the offline phase completes within minutes even for such sizes of $N$, which is acceptable given that policy updates occur infrequently (every 1000 observations in our experiments). In our deployment, policies are updated daily or weekly depending on traffic volume, providing ample time for offline computation. The batched nature of our algorithm naturally aligns with machine learning infrastructure designed for periodic model retraining.

While we have empirically evaluated two types of Bayesian models, our framework is compatible with any type of probabilistic model. For example, practitioners may utilize domain knowledge to develop grey-box models (see e.g., Schwartz et al. (2017)). Such models are simple to implement in probabilistic programming languages (Carpenter et al., 2017; Bingham et al., 2018; Tran et al., 2018), but challenging and inefficient to deploy. Our imitation framework can allow ease of deployment for these models while maintaining a comparable level of performance. Although we restricted attention to contextual bandits problems, an interesting research direction is to extend these methods to combinatorial ranking problems (Cheung et al., 2018; Dimakopoulou et al., 2019), where computational savings of distillation may be even larger.

While we focus on contextual bandits, our imitation framework naturally extends to sequential decision problems. The key challenge in reinforcement learning is that states evolve based on actions, introducing temporal dependencies that complicate off-policy learning. Several directions show promise: behavioral cloning with trajectory-level imitation where we imitate entire rollouts from a Thompson sampling agent; inverse reinforcement learning approaches that learn reward functions from TS demonstrations; generative adversarial imitation learning that matches state-action distributions without explicit reward modeling. The theoretical analysis would require extending our regret decomposition to handle state dependencies, potentially building on recent work connecting Thompson sampling to posterior sampling for reinforcement learning.

**Broader Impact Statement** Our work enables efficient deployment of Thompson sampling on resource-constrained devices, with demonstrated application in video upload systems serving millions of users. Our method allows sophisticated decision-making algorithms to run on low-end mobile devices common in developing economies, potentially reducing the digital divide. Better video quality and upload reliability

enhance communication, especially important for user-generated content platforms. By distilling complex policies into lightweight models, our approach may reduce energy consumption compared to running full Bayesian inference online.

On other other hand, contextual policies may perform differently across user subgroups. Users with older devices, slower connections, or in regions with poor infrastructure might systematically receive lower-quality recommendations. We mitigate this by including device and network features in the context, allowing the policy to adapt appropriately rather than disadvantaging certain users. Ongoing monitoring of performance across groups is essential.

Our method requires collecting contextual features from user devices. We emphasize that the deployed system operates on aggregated, anonymized features and does not require access to video content itself. The imitation learning phase uses logged features without personally identifiable information.

The reported watch time improvements raise broader questions about engagement-optimizing algorithms. While better video quality is generally beneficial, we acknowledge ongoing societal discussions about technology's role in attention capture.

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

# A    Imitation learning with Wasserstein distances

When actions can be naturally embedded in a continuous space, we may want to measure closeness between the imitation and TS policy by incorporating the geometry of the action space $\mathcal{A}$. In this section, we provide an alternative instantiation of the abstract form of Algorithm 1 by using Wasserstein distances as the notion of discrepancy $D\left(\cdot,\cdot\mid s\right)$. Our previous theoretical development for KL divergences has direct analogues in this setting, which we now briefly outline.

Given a metric $d(\cdot,\cdot)$ on $\mathcal{A}$, the Wasserstein distance between two distributions $q^1$ and $q^2$ on $\mathcal{A}$ is defined by the optimal transport problem

$$D_{\mathrm{w}}\left(q^1,q^2\right) = \inf_{\eta \in L(q^1,q^2)} \mathbb{E}_\eta[d(A,A')]$$

where $\eta(q^1,q^2)$ denotes the collection of all probabilities on $\mathcal{A} \times \mathcal{A}$ with marginals $q^1$ and $q^2$ (i.e., couplings). Intuitively, $D_{\mathrm{w}}\left(q^1,q^2\right)$ measures how much cost $d(A,A')$ is incurred by moving mass away from $A \sim q^1$ to $A' \sim q^2$ in an optimal fashion[5]. Wasserstein distances encode the geometry of the underlying space $\mathcal{A}$ via the distance $d$. Unlike the KL divergence $D_{\mathrm{kl}}\left(q^1\|q^2\right)$ that take value $\infty$ whenever $q^1$ has support not contained in $q^2$, Wasserstein distance allows imitation policies to have different support than the Thompson sampling policy, which is more appropriate in continuous action spaces. To simplify notation, for two policies $\pi^1$ and $\pi^2$, we let

$$D_{\mathrm{w}}\left(\pi^1,\pi^2\mid S\right) := D_{\mathrm{w}}\left(\pi^1(\cdot\mid S),\pi^2(\cdot\mid S)\right).$$

When Algorithm 1 is instantiated with the Wasserstein distance as its notion of discrepancy $D\left(\cdot,\cdot\mid S\right) := D_{\mathrm{w}}\left(\cdot,\cdot\mid S\right)$, the imitation learning problem equation 3 becomes

$$\underset{m\in\mathcal{M}}{\text{minimize}}\, \mathbb{E}_{S\sim\mathbb{P}_S}\left[D_{\mathrm{w}}\left(\bar{\pi},\pi^m\mid S\right)\right]. \tag{11}$$

To solve the above stochastic optimization problem, we can again use stochastic gradient descent methods, where the stochastic gradient $\nabla_m D_{\mathrm{w}}\left(\bar{\pi}_{\gamma(t)},\pi^m\mid S\right)$ can be computed by solving an optimal transport problem. From Kantorovich-Rubinstein duality (see, for example, Villani (2009)), we have

$$D_{\mathrm{w}}\left(\bar{\pi}_{\gamma(t)},\pi^m\mid s\right)$$
$$= \sup_{g:\mathcal{A}\to\mathbb{R}} \left\{\mathbb{E}_{\bar{A}\sim\bar{\pi}(\cdot|s)}g(\bar{A}) - \mathbb{E}_{A\sim\pi^m(\cdot|s)}g(A):\; g(a)-g(a')\le d(a,a')\text{ for all } a,a'\in\mathcal{A}\right\}, \tag{12}$$

where $d(\cdot,\cdot)$ is the metric on $\mathcal{A}$ used to define $D_{\mathrm{w}}\left(\cdot,\cdot\right)$. For discrete action spaces, the maximization problem equation 12 is a linear program with $O(|\mathcal{A}|)$ variables and constraints; for continuous action spaces, we can solve the problem over empirical distributions to approximate the optimal transport problem. We refer the interested reader to Peyré et al. (2019) for a comprehensive introduction to computational methods for solving optimal transport problems.

Letting $g^\star$ denote the optimal solution to the dual problem equation 12, the envelope theorem (or Danskin's theorem; Bonnans & Shapiro (2000, Theorem 4.13)) implies that under simple regularity conditions

$$\nabla_m D_{\mathrm{w}}\left(\bar{\pi}_{\gamma(t)},\pi^m\mid s\right) = -\nabla_m\mathbb{E}_{A\sim\pi^m(\cdot|s)}[g^\star(A)].$$

Assuming that an appropriate change of gradient and expectation is justified, we can use the policy gradient trick to arrive at

$$-\nabla_m\mathbb{E}_{A\sim\pi^m(\cdot|s)}[g^\star(A)] = -\mathbb{E}_{A\sim\pi^m(\cdot|s)}[g^\star(A)\nabla_m\log\pi^m(A\mid s)].$$

We conclude that for $A\sim\pi^m(\cdot\mid S_i)$,

$$-\,g^\star(A)\nabla_m\log\pi^m(A\mid S_i) \tag{13}$$

is a stochastic gradient for the imitation problem equation 11. As before, we can get lower variance estimates of the gradient by averaging the above estimator over many actions $A\sim\pi^m(\cdot\mid S_i)$. Using these stochastic gradients equation 13, we can solve the imitation problem equation 11 efficiently.

---

[5]For a discrete action space, $D_{\mathrm{w}}\left(\cdot,\cdot\right)$ can be defined with any symmetric matrix $d(a_i,a_j)$ satisfying $d(a_i,a_j)\ge 0$ with 0 iff $a_i = a_j$, and $d(a_i,a_j)\le d(a_i,a_k)+d(a_k,a_j)$ for any $a_i,a_j,a_k\in\mathcal{A}$.

We now show that the resulting imitation policy admits a regret decomposition similar to Lemma 1 for KL divergences. As a direct consequence of this decomposition, the regret bounds in Section 5.2 have their natural analogues with Wasserstein distances replacing KL divergences as the notion of discrepancy, though we omit them for brevity.

**Lemma 3.** *Let $U_t(\cdot; H_{\gamma(t)}, S_t) : \mathcal{A} \to \mathbb{R}$ be any upper confidence bound sequence that is measurable with respect to $H_{\gamma(t)}, S_t, A_t$. If there is a $L_\theta > 0$ satisfying almost surely*

$$|f_\theta(a, s) - f_\theta(a', s)| \le L_\theta d(a, a') \text{ for all } s \in \mathcal{S}, a, a' \in \mathcal{A}, \tag{14}$$

$$\text{BayesRegret}\left(T, \{\pi_{\gamma(t)}\}_{t \in \mathbb{N}}\right) \le \sum_{t=1}^{T} \mathbb{E}[f_\theta(A_t^\star, S_t) - U_t(A_t^\star; H_{\gamma(t)}, S_t)] + \sum_{t=1}^{T} \mathbb{E}[U_t(\bar{A}_t; H_{\gamma(t)}, S_t) - f_\theta(\bar{A}_t, S_t)]$$

$$+ \sum_{t=1}^{T} \mathbb{E}\left[L_\theta D_{\mathrm{w}}\left(\bar{\pi}_{\gamma(t)}, \pi_{\gamma(t)} \mid S_t\right)\right]. \tag{15}$$

*where $D_{\mathrm{w}}(\cdot, \cdot \mid \cdot)$ is the Wasserstein distance defined with the metric $d$ in the condition equation 14.*

**Proof** The proof mirrors that of Lemma 1. By the Kantorovich dual representation equation 12, we have

$$\mathbb{E}[f_\theta(\bar{A}_t, S_t) - f_\theta(A_t, S_t) \mid \theta, H_{\gamma(t)}, S_t] \le L_\theta D_{\mathrm{w}}\left(\bar{\pi}_{\gamma(t)}, \pi_{\gamma(t)} \mid S_t\right).$$

Here, we have again used that $\bar{A}_t \mid H_{\gamma(t)}, S_t$ and $A_t \mid H_{\gamma(t)}, S_t$ are independent of all else. Applying this bound in the decomposition equation 16, and taking expectation over $(H_{\gamma(t)}, S_t)$ on both sides and summing $t = 1, \ldots, T$, we get the desired bound. ⋄

# B Proof of Lemma 1

Conditional on $(H_{\gamma(t)}, S_t)$, $\bar{A}_t$ has the same distribution as $A_t^\star$. Since $U_t(a; H_{\gamma(t)}, S_t)$ is a deterministic function conditional on $(H_{\gamma(t)}, S_t)$, we have

$$\mathbb{E}[U_t(\bar{A}_t; H_{\gamma(t)}, S_t) \mid H_{\gamma(t)}, S_t] = \mathbb{E}[U_t(A_t^\star; H_{\gamma(t)}, S_t) \mid H_{\gamma(t)}, S_t].$$

We can rewrite the (conditional) instantenous regret as

$$\mathbb{E}[f_\theta(A_t^\star, S_t) - f_\theta(A_t, S_t) \mid H_{\gamma(t)}, S_t]$$
$$= \mathbb{E}[f_\theta(A_t^\star, S_t) - U_t(A_t^\star; H_{\gamma(t)}, S_t) \mid H_{\gamma(t)}, S_t] + \mathbb{E}[U_t(\bar{A}_t; H_{\gamma(t)}, S_t) - f_\theta(A_t, S_t) \mid H_{\gamma(t)}, S_t]$$
$$= \mathbb{E}[f_\theta(A_t^\star, S_t) - U_t(A_t^\star; H_{\gamma(t)}, S_t) \mid H_{\gamma(t)}, S_t] + \mathbb{E}[U_t(\bar{A}_t; H_{\gamma(t)}, S_t) - f_\theta(\bar{A}_t, S_t) \mid H_{\gamma(t)}, S_t]$$
$$+ \mathbb{E}[f_\theta(\bar{A}_t, S_t) - f_\theta(A_t, S_t) \mid H_{\gamma(t)}, S_t]. \tag{16}$$

We proceed by bounding the gap

$$\mathbb{E}[f_\theta(\bar{A}_t, S_t) - f_\theta(A_t, S_t) \mid \theta, H_{\gamma(t)}, S_t] \tag{17}$$

using the KL divergence between $\bar{\pi}_{\gamma(t)}$ and $\pi_{\gamma(t)}$. Recall Pinsker's inequality (Tsybakov, 2009)

$$\|P - Q\|_{\mathrm{TV}} := \sup_{g:\mathcal{A} \to [-1,1]} |\mathbb{E}_P[g(A)] - \mathbb{E}_Q[g(A)]| \le \sqrt{\frac{1}{2} D_{\mathrm{kl}}(P\|Q)}.$$

From the hypothesis, Pinsker's inequality implies

$$\mathbb{E}[f_\theta(\bar{A}_t, S_t) - f_\theta(A_t, S_t) \mid \theta, H_{\gamma(t)}, S_t] \le \sup_{a \in \mathcal{A}} |f_\theta(a, S_t)| \left\|\bar{\pi}_{\gamma(t)}(\cdot \mid S_t) - \pi_{\gamma(t)}(\cdot \mid S_t)\right\|_{\mathrm{TV}}$$

$$\le \sup_{a \in \mathcal{A}} |f_\theta(a, S_t)| \sqrt{\frac{1}{2} D_{\mathrm{kl}}\left(\bar{\pi}_{\gamma(t)}, \pi_{\gamma(t)} \mid S_t\right)}.$$

Here, we have used that $\bar{A}_t \mid H_{\gamma(t)}, S_t$ and $A_t \mid H_{\gamma(t)}, S_t$ are independent of all else.

Applying this bound in the decomposition equation 16, and taking expectation over $(H_{\gamma(t)}, S_t)$ on both sides and summing $t = 1, \ldots, T$, we get

$$\text{BayesRegret}\,(T, \pi) \leq \sum_{t=1}^{T} \mathbb{E}[f_\theta(A_t^\star, S_t) - U_t(A_t^\star; H_{\gamma(t)}, S_t)] + \sum_{t=1}^{T} \mathbb{E}[U_t(\bar{A}_t; H_{\gamma(t)}, S_t) - f_\theta(\bar{A}_t, S_t)]$$
$$+ \sum_{t=1}^{T} \mathbb{E}\left[\sup_{a \in \mathcal{A}} |f_\theta(a, S_t)| \sqrt{\frac{1}{2} D_{\mathrm{kl}}\left(\bar{\pi}_{\gamma(t)}, \pi_{\gamma(t)} \mid S_t\right)}\right].$$

Applying Cauchy-Schwarz inequality and noting that $\sqrt{\mathbb{E}[\sup_{a \in \mathcal{A}} f_\theta(a, S_t)^2]} \leq L$, we obtain the final decomposition. $\diamond$

## C Proof of regret bounds

### C.1 Proof of Theorem 1

In what follows, we abuse notation and let $C$ be a universal constant that changes line by line. We build on the batch UCB regret bound due to Desautels et al. (2014), defining the (batch) upper confidence bound

$$U_t(a; H_{\gamma(t)}, s) := \mu_{\gamma(t)}(a, s) + \sqrt{\beta_t} \sigma_{\gamma(t)}(a, s)$$

with $\beta_t = 2 \log((t^4 r d)^d t^2)$. From Borel-TIS inequality (e.g., see (Adler & Taylor, 2009)), we have

$$L^2 = \mathbb{E}\left[\sup_{a \in \mathcal{A}, s \in \mathcal{S}} f_\theta(a, s)^2\right] < \infty.$$

We bound the first two terms in the regret decomposition equation 5, starting with the second term

$$\sum_{t=1}^{T} \mathbb{E}[U_t(\bar{A}_t; H_{\gamma(t)}, S_t) - f_\theta(\bar{A}_t, S_t)] = \sum_{t=1}^{T} \sqrt{\beta_t} \mathbb{E}[\sigma_{\gamma(t)}(\bar{A}_t, S_t)].$$

First, note that since $|\sigma_{\gamma(t)}(a, s)| \leq \sqrt{c_2}$, Pinsker's inequality gives

$$|\mathbb{E}[\sigma_{\gamma(t)}(\bar{A}_t, S_t) \mid H_{\gamma(t)}, S_t] - \mathbb{E}[\sigma_{\gamma(t)}(A_t, S_t) \mid H_{\gamma(t)}, S_t]| \leq \sqrt{c_2} \left\|\bar{\pi}_{\gamma(t)}(\cdot \mid S_t) - \pi_{\gamma(t)}(\cdot \mid S_t)\right\|_{\mathrm{TV}}$$
$$\leq \sqrt{\frac{c_2}{2} D_{\mathrm{kl}}\left(\bar{\pi}_{\gamma(t)}, \pi_{\gamma(t)} \mid S_t\right)}.$$

We arrive at the interim bound

$$\sum_{t=1}^{T} \mathbb{E}[U_t(\bar{A}_t; H_{\gamma(t)}, S_t) - f_\theta(\bar{A}_t, S_t)]$$
$$\leq \sqrt{\beta_T} \sum_{t=1}^{T} \mathbb{E}[\sigma_{\gamma(t)}(A_t, S_t)] + \sqrt{\frac{\beta_T c_2}{2}} \sum_{t=1}^{T} \sqrt{\mathbb{E}[D_{\mathrm{kl}}\left(\bar{\pi}_{\gamma(t)}, \pi_{\gamma(t)} \mid S_t\right)]}. \tag{18}$$

By an elementary calculation (e.g., see Desautels et al. (2014, Proposition 1)), we have

$$\frac{\sigma_{\gamma(t)}(a, s)}{\sigma_t(a, s)} = \exp\left(I\left(f(s, a), \{R_s\}_{s=\gamma(t)}^{t-1} \mid \vec{R}_\gamma(t)\right)\right) \leq \sqrt{\eta_B},$$

where the last line follows from Assumption A. Next, we use the following lemma due to Srinivas et al. (2012).

**Lemma 4** (Srinivas et al. (2012, Lemma 5.3))**.** *For any sequence of $A_t$ and $S_t$,*

$$\mathbb{E}\left(\sum_{t=1}^{T}\sigma_t^2(A_t, S_t)\right)^{\frac{1}{2}} \leq \sqrt{\frac{2\gamma_T}{\log(1+\sigma^{-2})}}$$

Using the preceding two bounds, RHS of the inequality equation 18 can be further bounded by

$$\sum_{t=1}^{T}\mathbb{E}[U_t(\bar{A}_t; H_{\gamma(t)}, S_t) - f_\theta(\bar{A}_t, S_t)]$$

$$\leq \sqrt{T\eta_B\beta_T}\mathbb{E}\left[\left(\sum_{t=1}^{T}\sigma_t^2(A_t, S_t)\right)^{\frac{1}{2}}\right] + \sqrt{\frac{\beta_T c_2}{2}}\sum_{t=1}^{T}\sqrt{\mathbb{E}[D_{\mathrm{kl}}\left(\bar{\pi}_{\gamma(t)}, \pi_{\gamma(t)} \mid S_t\right)]}$$

$$\leq \sqrt{\frac{2T\eta_B\gamma_T\beta_T}{\log(1+\sigma^{-2})}} + \sqrt{\frac{\beta_T c_2}{2}}\sum_{t=1}^{T}\sqrt{\mathbb{E}[D_{\mathrm{kl}}\left(\bar{\pi}_{\gamma(t)}, \pi_{\gamma(t)} \mid S_t\right)]}. \tag{19}$$

We now bound the first term in the decomposition equation 5. Let $\mathcal{A}_t$ be a $(1/t^4)$-cover of $\mathcal{A}$, so that for any $a \in \mathcal{A}$, there exists $[a]_t \in \mathcal{A}_t$ such that $\|a - [a]_t\|_1 \leq 1/t^4$.

$$\sum_{t=1}^{T}\mathbb{E}[f_\theta(A_t^\star, S_t) - U_t(A_t^\star; H_{\gamma(t)}, S_t)] = \underbrace{\sum_{t=1}^{T}\mathbb{E}[f_\theta(A_t^\star, S_t) - f_\theta([A_t^\star]_t, S_t)]}_{(a)}$$

$$+ \underbrace{\sum_{t=1}^{T}\mathbb{E}[f_\theta([A_t^\star]_t, S_t) - U_t([A_t^\star]_t; H_{\gamma(t)}, S_t)]}_{(b)} + \underbrace{\sum_{t=1}^{T}\mathbb{E}[U_t([A_t^\star]_t; H_{\gamma(t)}, S_t) - U_t(A_t^\star; H_{\gamma(t)}, S_t)]}_{(c)}.$$

Using the definition of $L_f$, the first term $(a)$ in the above equality is bounded by

$$\sum_{t=1}^{T}\mathbb{E}[f_\theta(A_t^\star, S_t) - f_\theta([A_t^\star]_t, S_t)] \leq \mathbb{E}[L_f]\sum_{t=1}^{T}\|A_t^\star - [A_t^\star]_t\|_1 \leq \mathbb{E}[L_f]\sum_{t=1}^{\infty}\frac{1}{t^4} \leq C\mathbb{E}[L_f]$$

where we used the fact that $\mathcal{A}_t$ is a $1/t^4$-cover of $\mathcal{A}$.

To bound the second term $(b)$, use $f_\theta(a, s) \mid H_{\gamma(t)} \sim N(\mu_{\gamma(t)}(a, s), \sigma_{\gamma(t)}^2(a, s))$

$$\mathbb{E}[f_\theta(a, s) - U_t(a; H_{\gamma(t)}, s) \mid H_{\gamma(t)}] \leq \mathbb{E}[\left(f_\theta(a, s) - U_t(a; H_{\gamma(t)}, s)\right)_+ \mid H_{\gamma(t)}]$$

$$= \frac{\sigma_{\gamma(t)}(a, s)}{\sqrt{2\pi}}e^{-\frac{\beta_t}{2}} \leq \frac{c_2}{\sqrt{2\pi}t^2|\mathcal{A}_t|}, \tag{20}$$

where we used $2\log(|\mathcal{A}_t|t^2) \leq \beta_t$ since $|\mathcal{A}_t| \leq (t^4 rd)^d$. Hence, we obtain the bound

$$\sum_{t=1}^{T}\mathbb{E}[f_\theta([A_t^\star]_t, S_t) - U_t([A_t^\star]_t; H_{\gamma(t)}, S_t)] \leq \sum_{t=1}^{T}\sum_{a \in \mathcal{A}_t}\mathbb{E}[f_\theta(a, S_t) - U_t(a; H_{\gamma(t)}, S_t)] \leq \sum_{t=1}^{\infty}\frac{c_2}{\sqrt{2\pi}t^2} \leq Cc_2$$

where we used the independence of $S_t$ and $H_{\gamma(t)}$, and the bound equation 20.

To bound the third term $(c)$, we show the claim

$$|U_t(a; H_{\gamma(t)}, s) - U_t(a'; H_{\gamma(t)}, s)| \leq \mathbb{E}[L_f \mid H_{\gamma(t)}]\|a - a'\|_1 \tag{21}$$

$$+ \sqrt{\beta_t}\left(2\mathbb{E}\left[L_f\left(\sup_{a \in \mathcal{A}, s \in \mathcal{S}}\mu(a, s)^2 + \sup_{a \in \mathcal{A}, s \in \mathcal{S}}f_\theta(a, s)^2\right) \mid H_{\gamma(t)}\right]\right)^{\frac{1}{2}}\|a - a'\|_1^{\frac{1}{2}}.$$

From the above claimed bound, it follows that

$$\sum_{t=1}^{T} \mathbb{E}[U_t([A_t^\star]_t; H_{\gamma(t)}, S_t) - U_t(A_t^\star; H_{\gamma(t)}, S_t)] \leq \sum_{t=1}^{T} \frac{\mathbb{E}[L_f]}{t^4} + \sum_{t=1}^{T} \sqrt{2\beta_t} \frac{c_1\sqrt{\mathbb{E}[L_f]} + c_3\sqrt{\mathbb{E}[L_f^2]}}{t^2}$$

$$\leq C\mathbb{E}[L_f] + Cd\log(rd)\left(c_1\sqrt{\mathbb{E}[L_f]} + c_3\sqrt{\mathbb{E}[L_f^2]}\right).$$

To show the bound equation 21, first note $a \mapsto \mathbb{E}[f_\theta(a,s) \mid H_{\gamma(t)}]$ and $a \mapsto \mathbb{E}[f_\theta(a,s)^2 \mid H_{\gamma(t)}]$ are $\mathbb{E}[L_f \mid H_{\gamma(t)}]$- and $\mathbb{E}[2L_f \sup_{a\in\mathcal{A},s\in\mathcal{S}} |f_\theta(a,s)| \mid H_{\gamma(t)}]$- Lipschitz respectively, for all $s \in \mathcal{S}$. Hence, $a \mapsto \sigma_{\gamma(t)}^2(a,s)$ is $\mathbb{E}[2L_f(c_1^2 + \sup_{a\in\mathcal{A},s\in\mathcal{S}} |f_\theta(a,s)|^2) \mid H_{\gamma(t)}]$-Lipschitz. Noting that

$$|\sigma_{\gamma(t)}(a,s) - \sigma_{\gamma(t)}(a',s)| = \left|\frac{\sigma_{\gamma(t)}^2(a,s) - \sigma_{\gamma(t)}^2(a',s)}{\sigma_{\gamma(t)}(a,s) + \sigma_{\gamma(t)}(a',s)}\right| \leq \frac{1}{c}|\sigma_{\gamma(t)}^2(a,s) - \sigma_{\gamma(t)}^2(a',s)| + c$$

for any $c > 0$, taking the infimum over $c > 0$ on the right hand side yields

$$|\sigma_{\gamma(t)}(a,s) - \sigma_{\gamma(t)}(a',s)| \leq \sqrt{2|\sigma_{\gamma(t)}^2(a,s) - \sigma_{\gamma(t)}^2(a',s)|}$$

$$\leq \left(2\mathbb{E}\left[L_f\left(c_1^2 + \sup_{a\in\mathcal{A},s\in\mathcal{S}} f_\theta(a,s)^2\right) \mid H_{\gamma(t)}\right]\right)^{\frac{1}{2}} \|a - a'\|_1^{\frac{1}{2}}$$

which shows the bound equation 21.

Collecting these bounds, we have shown that

$$\sum_{t=1}^{T} \mathbb{E}[f_\theta(A_t^\star, S_t) - U_t(A_t^\star; H_{\gamma(t)}, S_t)] \leq C\mathbb{E}[L_f] + Cc_2 + Cd\log(rd)\left(c_1\sqrt{\mathbb{E}[L_f]} + c_3\sqrt{\mathbb{E}[L_f^2]}\right). \tag{22}$$

Combining this with the bound equation 19, we obtain our result.

## D  Proof of generalization results

### D.1  Proof of Theorem 3

We abuse notation and use $C > 0$ to denote a numerical constant that changes value line to line. We use the following concentration guarantee using localized Rademacher averages.

**Lemma 5** (Bartlett et al. (2005, Theorem 3.3)). *For a class of functions $\mathcal{G}$ with range $[0, M]$, let $r_n^\star$ be the unique positive fixed point of the sub-root function $\psi_n$ satisfying the bound equation 10. Then, for i.i.d. observations $\xi \overset{\text{iid}}{\sim} \mathbb{P}$, there is a numerical constant $C > 0$ such that*

$$\mathbb{E}[g] \leq \left(1 + \frac{1}{\eta}\right)\frac{1}{n}\sum_{i=1}^{n} g(\xi_i) + C(1+\eta)\left(\frac{1}{M}r_n^\star + \frac{Mz}{n}\right) + \frac{CMz}{n} \quad \text{for all } g \in \mathcal{G} \text{ and } \eta \geq 0$$

*with probability at least $1 - e^{-z}$.*

Notice that by Jensen inequality, we have

$$\mathbb{E}_{\bar{A}\sim\bar{\pi}(\cdot|S_i)}\left[\log\frac{\bar{\pi}(\bar{A} \mid S_i)}{\widehat{\pi^m}(\bar{A} \mid S_i)}\right] \geq 0 \quad \text{almost surely.}$$

Applying Lemma 5 with the function class $\mathcal{G}_1$ and $\eta = 1/2$, we have

$$\mathbb{E}\left[D_{\mathrm{kl}}\left(\bar{\pi}, \widehat{\pi^m} \mid S\right)\right] \leq \frac{3}{2}\frac{1}{N}\sum_{i=1}^{N}\mathbb{E}_{\bar{A}\sim\bar{\pi}(\cdot|S_i)}\left[\log\frac{\bar{\pi}(\bar{A} \mid S_i)}{\widehat{\pi^m}(\bar{A} \mid S_i)}\right] + \frac{C}{M}r_N^\star + \frac{CMz}{N}. \tag{23}$$

In the rest of the proof, we bound the interim (uniform) approximation error

$$Z_{N,N_a} := \sup_{m \in \mathcal{M}} \left\{ \frac{1}{N} \sum_{i=1}^{N} \left( \mathbb{E}_{\bar{A} \sim \bar{\pi}(\cdot|S_i)} \left[ \log \frac{\bar{\pi}(\bar{A} \mid S_i)}{\pi^m(\bar{A} \mid S_i)} \right] - \frac{1}{N_a} \sum_{j=1}^{N_a} \log \frac{\bar{\pi}(\bar{A}_{ij} \mid S_i)}{\pi^m(\bar{A}_{ij} \mid S_i)} \right) \right\}.$$

This is indeed sufficient for our purposes since the bound equation 23 implies

$$\mathbb{E}\left[ D_{\mathrm{kl}}\left( \bar{\pi}, \pi^{\widehat{m}} \mid S \right) \right] \leq \frac{3}{2} \frac{1}{N} \sum_{i=1}^{N} \frac{1}{N_a} \sum_{j=1}^{N_a} \log \frac{\bar{\pi}(\bar{A}_{ij} \mid S_i)}{\pi^{\widehat{m}}(\bar{A}_{ij} \mid S_i)} + C \left( Z_{N,N_a} + \frac{1}{M} r_N^{\star} + \frac{Mz}{N} \right).$$

By the definition equation 8 of the empirical solution $\widehat{m}$ and by virtue of having a well-specified model class $\mathcal{M}$, the first term in the preceding bound is nonpositive

$$\frac{1}{N} \sum_{i=1}^{N} \frac{1}{N_a} \sum_{j=1}^{N_a} \log \frac{\bar{\pi}(\bar{A}_{ij} \mid S_i)}{\pi^{\widehat{m}}(\bar{A}_{ij} \mid S_i)} \leq \frac{1}{N} \sum_{i=1}^{N} \frac{1}{N_a} \sum_{j=1}^{N_a} \log \frac{\bar{\pi}(\bar{A}_{ij} \mid S_i)}{\bar{\pi}^{m^{\star}}(\bar{A}_{ij} \mid S_i)} = 0.$$

Consider the Doob martingale $M_0 = \mathbb{E}[Z_{N,N_a}]$, and

$$M_k := \mathbb{E}[Z_{N,N_a} \mid S_1, \ldots, S_k] \quad \text{for} \quad 1 \leq k \leq N,$$

a martingale adapted to the filtration $\mathcal{F}_k := \sigma(S_1, \ldots, S_k)$. Denote the martingale difference sequence $D_k = M_k - M_{k-1}$ for $k \geq 1$. Let $S_k'$ be an independent copy of $S_k$ that is independent of all $S_i$, and let $\bar{A}_{kj}' \sim \bar{\pi}(\cdot \mid S_k')$ independent of everything other than $S_k'$. We can write

$$D_k = \mathbb{E}\left[ \sup_{m \in \mathcal{M}} \left\{ \frac{1}{N} \sum_{i=1}^{N} \left( \mathbb{E}_{\bar{A} \sim \bar{\pi}(\cdot|S_i)} \left[ \log \frac{\bar{\pi}(\bar{A} \mid S_i)}{\pi^m(\bar{A} \mid S_i)} \right] - \frac{1}{N_a} \sum_{j=1}^{N_a} \log \frac{\bar{\pi}(\bar{A}_{ij} \mid S_i)}{\pi^m(\bar{A}_{ij} \mid S_i)} \right) \right\} \mid S_1, \ldots, S_k \right]$$

$$- \mathbb{E}\left[ \sup_{m \in \mathcal{M}} \left\{ \frac{1}{N} \sum_{i \neq k} \left( \mathbb{E}_{\bar{A} \sim \bar{\pi}(\cdot|S_i)} \left[ \log \frac{\bar{\pi}(\bar{A} \mid S_i)}{\pi^m(\bar{A} \mid S_i)} \right] - \frac{1}{N_a} \sum_{j=1}^{N_a} \log \frac{\bar{\pi}(\bar{A}_{ij} \mid S_i)}{\pi^m(\bar{A}_{ij} \mid S_i)} \right) \right. \right.$$

$$\left. \left. + \frac{1}{N} \left( \mathbb{E}_{\bar{A}' \sim \bar{\pi}(\cdot|S_k')} \left[ \log \frac{\bar{\pi}(\bar{A}' \mid S_k')}{\pi^m(\bar{A}' \mid S_k')} \right] - \frac{1}{N_a} \sum_{j=1}^{N_a} \log \frac{\bar{\pi}(\bar{A}_{ij}' \mid S_k')}{\pi^m(\bar{A}_{ij}' \mid S_k')} \right) \right\} \middle| S_1, \ldots, S_k \right].$$

Independence of $S_i$'s yields

$$|D_k| \leq \frac{1}{N} \mathbb{E}\left[ \sup_{m \in \mathcal{M}} \left| \frac{1}{N_a} \sum_{j=1}^{N_a} \mathbb{E}_{\bar{A} \sim \bar{\pi}(\cdot|S_k)} \left[ \log \frac{\bar{\pi}(\bar{A} \mid S_k)}{\pi^m(\bar{A} \mid S_k)} \right] - \log \frac{\bar{\pi}(\bar{A}_{ij} \mid S_k)}{\pi^m(\bar{A}_{ij} \mid S_k)} \right. \right.$$

$$\left. \left. - \mathbb{E}_{\bar{A}' \sim \bar{\pi}(\cdot|S_k')} \left[ \log \frac{\bar{\pi}(\bar{A}' \mid S_k')}{\pi^m(\bar{A}' \mid S_k')} \right] + \log \frac{\bar{\pi}(\bar{A}_{ij}' \mid S_k')}{\pi^m(\bar{A}_{ij}' \mid S_k')} \right| \middle| S_k \right]$$

$$\leq \frac{2}{N} \sup_{s \in \mathcal{S}} \mathbb{E}_{\bar{A}_j \overset{\mathrm{iid}}{\sim} \bar{\pi}(\cdot|s)} \left[ \sup_{m \in \mathcal{M}} \left| \frac{1}{N_a} \sum_{j=1}^{N_a} \mathbb{E}_{\bar{A} \sim \bar{\pi}(\cdot|s)} \left[ \log \frac{\bar{\pi}(\bar{A} \mid s)}{\pi^m(\bar{A} \mid s)} \right] - \log \frac{\bar{\pi}(\bar{A}_j \mid s)}{\pi^m(\bar{A}_j \mid s)} \right| \right].$$

Next, we use a standard symmetrization result to bound the last display; see, for example, Chapter 2.3, van der Vaart & Wellner (1996) for a comprehensive treatment.

**Lemma 6.** *If $\xi_i \overset{\mathrm{iid}}{\sim} P$, we have*

$$\mathbb{E}\left[ \sup_{g \in \mathcal{G}} \left| \frac{1}{n} \sum_{i=1}^{n} (g(\xi_i) - \mathbb{E}[g(\xi)]) \right| \right] \leq 4\mathbb{E}[\mathfrak{R}_n(\mathcal{G})]$$

Applying Lemma 6 to the bound on $|D_k|$. we conclude $|D_k| \leq \frac{8}{N} \sup_{s \in \mathcal{S}} \mathbb{E}_{\bar{A}_j \overset{\text{iid}}{\sim} \bar{\pi}(\cdot|s)}[\mathfrak{R}_{N_a}(\mathcal{G}'_2(s))]$, where $\mathcal{G}'_2(s)$ is the function class

$$\mathcal{G}'_2(s) := \left\{ a \mapsto \log \frac{\bar{\pi}(a \mid s)}{\pi^m(a \mid s)} : m \in \mathcal{M} \right\}.$$

Note that $\mathfrak{R}_{N_a}(\mathcal{G}'_2(s)) = \mathfrak{R}_{N_a}(\mathcal{G}_2(s))$. Then, Azuma-Hoeffding bound (Corollary 2.20, Wainwright (2019)) yields

$$Z_{N,N_a} \leq \mathbb{E}[Z_{N,N_a}] + \sqrt{\frac{32z}{N}} \sup_{s \in \mathcal{S}} \mathbb{E}_{\bar{A}_j \overset{\text{iid}}{\sim} \bar{\pi}(\cdot|s)}[\mathfrak{R}_{N_a}(\mathcal{G}_2(s))]$$

with probability at least $1 - e^{-z}$.

It now remains to bound $\mathbb{E}[Z_{N,N_a}]$, for which we use a symmetrization argument. Although $(S_i, \bar{A}_{ij})$ are not i.i.d., a standard argument still applies, which we outline for completeness. Denoting by $(S'_i, \bar{A}'_{ij})$ independent copies of $(S_i, \bar{A}_{ij})$ again, we have

$$\mathbb{E}[Z_{N,N_a}] = \mathbb{E}\left[ \sup_{m \in \mathcal{M}} \left| \frac{1}{N}\sum_{i=1}^{N} \frac{1}{N_a}\sum_{j=1}^{N_a} \log \frac{\bar{\pi}(\bar{A}_{ij} \mid S_i)}{\pi^m(\bar{A}_{ij} \mid S_i)} - \mathbb{E}\left[ \frac{1}{N}\sum_{i=1}^{N} \frac{1}{N_a}\sum_{j=1}^{N_a} \log \frac{\bar{\pi}(\bar{A}'_{ij} \mid S'_i)}{\pi^m(\bar{A}'_{ij} \mid S'_i)} \right] \right| \right]$$

$$\leq \mathbb{E}\left[ \sup_{m \in \mathcal{M}} \left| \frac{1}{N}\sum_{i=1}^{N} \frac{1}{N_a}\sum_{j=1}^{N_a} \log \frac{\bar{\pi}(\bar{A}_{ij} \mid S_i)}{\pi^m(\bar{A}_{ij} \mid S_i)} - \log \frac{\bar{\pi}(\bar{A}'_{ij} \mid S'_i)}{\pi^m(\bar{A}'_{ij} \mid S'_i)} \right| \right]$$

$$= \mathbb{E}\left[ \sup_{m \in \mathcal{M}} \left| \frac{1}{N}\sum_{i=1}^{N} \frac{1}{N_a}\sum_{j=1}^{N_a} \epsilon_{ij}\left( \log \frac{\bar{\pi}(\bar{A}_{ij} \mid S_i)}{\pi^m(\bar{A}_{ij} \mid S_i)} - \log \frac{\bar{\pi}(\bar{A}'_{ij} \mid S'_i)}{\pi^m(\bar{A}'_{ij} \mid S'_i)} \right) \right| \right]$$

$$\leq 2\mathbb{E}[\mathfrak{R}_{NN_a}(\mathcal{G}_3)].$$

Collecting these bounds, we obtain the desired result.

## D.2 Proof of Corollary 1

We use the following standard result that bound the Rademacher complexity of kernel models. Let $k$ be a reproducing kernel on $\Xi$, and let $\mathbb{B}$ be the unit ball in the RKHS $\mathcal{H}$.

**Claim 7.** *Let* $\sup_{\xi \in \Xi} k(\xi, \xi) = B < \infty$. *Then,* $\mathfrak{R}_n(\mathbb{B}) \leq \frac{B}{\sqrt{n}}$.

**Proof of Claim**  For any fixed $\xi_1, \ldots, \xi_n$,

$$\mathfrak{R}_n(\mathbb{B}) = \frac{1}{n}\mathbb{E}_\varepsilon\left[ \sup_{h \in \mathbb{B}} \left\langle h, \sum_{i=1}^{n} \varepsilon_i k(\cdot, \xi_i) \right\rangle \right] = \frac{1}{n}\mathbb{E}_\varepsilon\left[ \left\| \sum_{i=1}^{n} \varepsilon_i k(\cdot, \xi_i) \right\|_{\mathcal{H}} \right]$$

$$\leq \frac{1}{n}\left( \mathbb{E}_\varepsilon\left[ \left\| \sum_{i=1}^{n} \varepsilon_i k(\cdot, \xi_i) \right\|_{\mathcal{H}}^2 \right] \right)^{\frac{1}{2}} = \frac{1}{n}\sqrt{\sum_{i=1}^{n} \|k(\cdot, \xi_i)\|_{\mathcal{H}}^2} \leq \frac{B}{\sqrt{n}}.$$

$\square$

Applying the claim to $\mathbb{B}_{\mathcal{A}}$ and and $\mathbb{B}_{\mathcal{S} \times \mathcal{A}}$, we get

$$\sup_{s \in \mathcal{S}} \mathfrak{R}_{N_a}(\mathcal{G}_2(s)) \leq \frac{B}{\sqrt{N_a}} \quad \text{and} \quad \mathfrak{R}_{NN_a}(\mathcal{G}_3) \leq \frac{B}{\sqrt{NN_a}}.$$

To bound $r_N^\star$, we use the following result due to Mendelson (2003).

**Lemma 8** (Mendelson (2003, Theorem 2.1)). *If $\lambda_1 \geq 1/N$, then for all $r \geq 1/N$*

$$\mathbb{E}\left[\mathfrak{R}_N\left\{h \in \mathbb{B} : \mathbb{E}[h(S)^2] \leq r\right\}\right] \lesssim \left(\frac{1}{N}\sum_{j=1}^{\infty}\min\{\lambda_j, r\}\right)^{\frac{1}{2}}.$$

Consider the case where the spectrum of $T_{k_S}$ decay exponentially

$$\left(\frac{1}{N}\sum_{j=1}^{\infty}\min\left\{e^{-j^2}, \frac{\sqrt{\log N}}{N}\right\}\right)^{\frac{1}{2}} \lesssim \left(\frac{1}{N}\sum_{j=1}^{\sqrt{\log N}}\frac{\sqrt{\log N}}{N} + \frac{1}{N}\int_{\sqrt{\log N}}^{\infty}e^{-t^2}dt\right)^{\frac{1}{2}} \lesssim \frac{\sqrt{\log N}}{N},$$

where we use $\lesssim$ to denote inequality up to a numerical constant. We conclude $r_N^\star \lesssim M\frac{\sqrt{\log N}}{N}$. For polynomially decaying spectrum $\lambda_j \lesssim j^{-2\beta}$,

$$\sum_{j=1}^{\infty}\min\{j^{-2\beta}, r\} \approx r^{\frac{2\beta-1}{2\beta}} + \int_{r^{-1/2\beta}}^{\infty}t^{-2\beta}dt \asymp r^{\frac{2\beta-1}{2\beta}}.$$

Solving for the fixed point, we get $r_N^\star \asymp Mn^{-\frac{2\beta}{2\beta+1}}$.

Collecting these bounds and plugging them into Theorem 3, we obtain the desired result.

## E   Experiment Details

**Hyperparameters**   We use hyperparameters from Riquelme et al. (2018) as follows. The NEURALGREEDY, NEURALLINEARTS methods use a fully-connected neural network with two hidden layers of containing 100 rectified linear units. The networks are multi-output, where each output corresponds for predicted reward under each action. The networks are trained using 100 mini-batch updates at each period to minimize the mean-squared error via RMSProp with an initial learning rate of 0.01. The learning rate is decayed after each mini-batch update according to an inverse time decay schedule with a decay rate of 0.55 and the learning rate is reset the initial learning rate each update period. For BOOTSTRAP-NN-TS, we use 10 replicates and train each replicate with all observations as in Riquelme et al. (2018).

The Bayesian linear regression models used on the last linear layer for NEURALLINEAR-TS use the normal inverse gamma prior $\mathrm{NIG}(\mu_a = \mathbf{0}, \alpha_a = 3, \beta_a = 3, \Lambda_a = 0.25I_d)$. LINEAR-TS uses a $\mathrm{NIG}(\mu_a = \mathbf{0}, \alpha_a = 6, \beta_a = 6, \Lambda_a = 0.25I_d)$ prior distribution.

The imitation models used by the IL methods are fully-connected neural networks with two hidden layers of 100 units and hyperbolic tangent activations. The networks use a Softmax function on the outputs to predict the probability of selecting each action. The networks are trained using 2000 mini-batch updates via RMSProp to minimize the KL-divergence between the predicted probabilities and the approximate propensity scores of the Thompson sampling policy $\pi^{TS}$. For each observed context $S_i$, we approximate the propensity scores of the Thompson sampling policy $\pi^{TS}(\cdot|S_i)$ using $N_a = 2048$ Monte Carlo samples: $\hat{\pi}^{TS}(a|S_i) = \frac{1}{N_a}\sum_{j=1}^{N_a}\mathbf{1}\{A_{ij} = a\}$ where $A_{ij} \sim \pi^{TS}(\cdot|S_i)$. We use an initial learning rate of 0.001. learning rate is decayed every 100 mini-batches according to an inverse time decay schedule with a decay rate of 0.05. In practice, the hyperparameters of the imitation model can be optimized or adjusted at each update period by minimizing the KL-divergence on a held-out subset of the observed data, which may lead to better regret performance. We do not use inverse propensity-weighting on the observations, but we suspect that may it may further improve performance.

We normalize all numeric features to be in [0,1] and one-hot encode all categorical features. For the Warfarin dataset, we also normalize the rewards to be in [0,1].

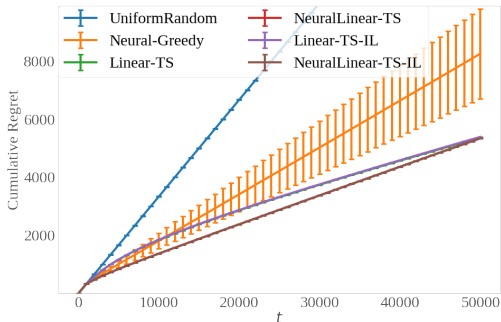

Figure 3: Cumulative regret on the Warfarin problem with 50 actions

**Posterior Inference for Bayesian Linear Regression   Linear-TS**: For each action, We assume the data for action $a$ were generated from the linear function: $r_a = \boldsymbol{s}^T \boldsymbol{\theta}_a + \varepsilon$ where $\varepsilon \sim \mathcal{N}(0, \sigma_a^2)$.

$$\sigma_a^2 \sim \text{IG}(\alpha_a, \beta_a), \quad \boldsymbol{\theta_a}|\sigma_a^2 \sim \mathcal{N}(\boldsymbol{\mu}_a, \sigma_a^2 \Sigma_a),$$

where the prior distribution is given by $\text{NIG}(\boldsymbol{\mu}_a, \Lambda_a, \alpha_a, \beta_a)$ and $\Lambda_a = \Sigma_a^{-1}$ is the precision matrix. After $n_a$ observations of contexts $X_a \in \mathbb{R}^{n_a \times (d+1)}$ and rewards $\boldsymbol{y}_a \in \mathbb{R}^{n_a \times 1}$, we denote the joint posterior by $P(\boldsymbol{\theta}_a, \sigma_a^2) \sim \text{NIG}(\bar{\boldsymbol{\mu}}_a, \bar{\Lambda}_a, \bar{\alpha}_a, \bar{\beta}_a)$, where

$$\bar{\Lambda} = X_a^T X_a + \Lambda_a, \quad \bar{\boldsymbol{\mu}}_a = \bar{\Lambda}_a^{-1}(\Lambda_a \boldsymbol{\mu}_a + X_a^T \boldsymbol{y}_a)$$

$$\bar{\alpha}_a = \alpha + \frac{n_a}{2}, \quad \bar{\beta}_a = \beta + \frac{1}{2}(\boldsymbol{y}_a^T \boldsymbol{y}_a + \boldsymbol{\mu}_a^T \Lambda_a \boldsymbol{\mu}_a - \bar{\boldsymbol{\mu}}_a^T \bar{\Lambda}_a \bar{\boldsymbol{\mu}}_a).$$

**Additional Results   Warfarin - 50 Actions** Figure 3 shows the cumulative regret on Warfarin using 50 actions. The imitation learning methods match the cumulative regret of the vanilla Thompson sampling methods.

## F   Time and Space Complexity

### F.1   Complexity of Evaluated Methods

Table 2 shows the decision-making time complexity for the methods used in our empirical analysis. The time complexity is equivalent to the space complexity for all evaluated methods.

**NeuralGreedy** The time complexity of NEURALGREEDY is the sum of matrix-vector multiplications involved in a forward pass.

**Linear-TS** The time complexity of LINEAR-TS is dominated by sampling from the joint posterior, which requires sampling from a multivariate normal with dimension $d$. To draw a sample from the joint posterior $P(\boldsymbol{\theta}, \sigma)$ at decision time, we first sample the noise level $\tilde{\sigma}^2 \sim \text{IG}(\alpha, \beta)$ and then sample $\tilde{\boldsymbol{\theta}}|\tilde{\sigma}^2 \sim \mathcal{N}(\boldsymbol{\mu}, \tilde{\sigma}^2 \Lambda^{-1})$. Rather than inverting the precision matrix $\tilde{\Sigma} = \tilde{\sigma}^2 \Lambda^{-1}$, we compute root decomposition (e.g. a Cholesky decomposition) of the $d \times d$ precision matrix $\Lambda = LL^T$. The root decomposition can be computed once, with cost $O(d^3)$, after an offline batch update and cached until the next batch update. Given $L^T$, we sample directly by computing $\tilde{\boldsymbol{\theta}} = \boldsymbol{\mu} + \boldsymbol{z}$, where

$$\frac{1}{\tilde{\sigma}} L^T \boldsymbol{z} = \boldsymbol{\zeta} \tag{24}$$

and $\boldsymbol{\zeta} \overset{\text{iid}}{\sim} \mathcal{N}(0, 1)$. Since $L^T$ is upper triangular, Eqn. equation 24 can be solved using a backward substitution in quadratic time: $O(d^2)$.[6]

---

[6]The alternative approach of inverting the precision matrix to compute the covariance matrix $\Sigma = \Lambda^{-1}$, computing and caching its root decomposition $\Sigma = L_\Sigma L_\Sigma^T$, and sampling $\tilde{\boldsymbol{\theta}}$ as $\tilde{\boldsymbol{\theta}} = \boldsymbol{\mu} + L_\Sigma \boldsymbol{\zeta}$, where $\boldsymbol{\zeta} \overset{\text{iid}}{\sim} \mathcal{N}(0, 1)$ also has a time complexity of $O(d^2)$ from the matrix-vector multiplication $L_\Sigma \boldsymbol{\zeta}$.

**NeuralLinear-TS** The time complexity of NEURALLINEAR-TS is the sum of a forward pass up to the last hidden layer and sampling from a multivariate normal with dimension $h_M$, where $h_M$ is the size of the last hidden layer.

**Imitation Learning** The IL methods have the same time complexity as NEURALGREEDY, ignoring the cost of sampling from multinomial with $k$ categories.

### F.2 Complexity Using Embedded Actions

An alternative modeling approach for the non-imitation methods is to embed the action with the context as input to the reward model.

**NeuralGreedy** Using an embedded action, the time complexity for a forward pass up to the last layer is $O_{\text{last-layer}} = O\big(kd_a h_1 + k \sum_{m=1}^{M-1} h_m h_{m+1}\big)$ because the input at decision time is a $k \times d_a$ matrix where the context is embedded with each of the $k$ actions and the each context-action vector has dimension $d_a$. The time complexity of computing the output layer remains $O(kh_M)$. The space complexity remains linear in the number of parameters, but it also requires computing temporary intermediate tensors of size $k \times h_m$ for $m = 1...M$: $O\big(d_a h_1 + \sum_{m=1}^{M-1} h_m h_{m+1} + \sum_{m=1}^{M} kh_m\big)$.

**Linear-TS** Linear-TS with an embedded action only requires using a single sample of the parameters, which yields a complexity of to $O(d_a^2 + kd_a)$ for LINEAR-TS. The space complexity is also $O(d_a^2 + kd_a)$.

**NeuralLinear-TS** For NEURALLINEAR-TS the time complexity of computing the outputs given the last hidden layer is $O(h_M^2 + kh_M)$, since only a single sample of $h_M$ parameters is required for computed the reward for all actions. The space complexity for NEURALLINEAR-TS the sum the space complexities of NEURALGREEDY and LINEAR-TS.

**Imitation Learning** The computatiuonal cost of the IL methods would be unchanged.

We choose to empirically evaluate models *without* embedded actions because linear methods using embedded actions cannot model reward functions that involve non-linear interactions between the contexts and actions, whereas modeling each action independently allows for more flexibility. Riquelme et al. (2018) find that Thompson sampling using disjoint, exact linear bayesian regressions are a strong baseline in many applications. Furthermore, Riquelme et al. (2018) observe that it is important to model the noise levels independently for each action.

### F.3 Complexity of Alternative Methods

Alternative Thompson sampling methods including mean-field approaches, the low-rank approximations of the covariance matrix, and bootstrapping can also decrease the computational cost of posterior sampling. Mean-field approaches can reduce time complexity of sampling parameters from the posterior from quadratic $O(n^2)$ to linear $O(n)$ in the number of parameters $n$.[7] However, assuming independence among parameters has been observed to result in worse performance in some settings (Riquelme et al., 2018). Low-rank approximations of the covariance matrix allow for sampling parameters in $O((n+1)\rho)$, where $\rho$ is the rank of the approximate covariance, but such methods have a space complexity of $O(\rho n)$ since they require storing $\rho$ copies of the parameters Zhang et al. (2018); Maddox et al. (2019). Bootstrapping also requires storing multiple copies of the parameters, so the space is $O(bn)$ where $b$ is the number of bootstrap replicates. However, bootstrapping simply requires a multinomial draw to select one set of bootstrapped parameters. All these methods require a forward pass using the sampled parameters, and the time complexity is the sum of the time complexities of sampling parameters and the forward pass.

---

[7]We describe space complexity in terms of the number of parameters $n$, so that we do not make assumptions about the underlying model.

Table 2: Decision-making time complexity and space complexity for each method . For methods relying on fully-connected neural networks, the time complexity of a forward pass to the last hidden layer is $C_{\text{last-layer}} = dh_1 + \sum_{m=1}^{M-1} h_m h_{m+1}$, where $d$ is the dimension of the context and $h_m$ is the number of units in hidden layer $m$. For BOOTSTRAP-NN-TS, $B$ denotes the number of bootstrap replicates.

| METHOD | TIME COMPLEXITY | SPACE COMPLEXITY |
|---|---|---|
| NEURALGREEDY | $O(C_{\text{LAST-LAYER}}) + O(kh_M)$ | $O(C_{\text{LAST-LAYER}}) + O(kh_M)$ |
| LINEAR-TS | $O(kd^2)$ | $O(kd^2)$ |
| NEURALLINEAR-TS | $O(C_{\text{LAST-LAYER}}) + O\left(kh_M^2\right)$ | $O(C_{\text{LAST-LAYER}}) + O\left(kh_M^2\right)$ |
| BOOTSTRAP-NN-TS | $O(C_{\text{LAST-LAYER}}) + O(kh_M)$ | $O(C_{\text{LAST-LAYER}} \cdot B) + O(kh_M B))$ |
| IL | $O(C_{\text{LAST-LAYER}}) + O(kh_M)$ | |

