# OpenReview forum: "Distilled Thompson Sampling: Practical and Efficient Thompson Sampling via Imitation Learning"
_TMLR — Accepted by TMLR_

### Review · Reviewer_SjvG · 2025-11-18

**Summary Of Contributions:**

The paper proposes a distilled version of Thompson Sampling (TS), designed to deploy contextual bandit algorithms in low-latency, resource-constrained environments (e.g., mobile devices) leveraging imitation learning. The core method involves an architecture made up of:

- Offline TS policy: A standard TS policy runs on backend servers, performing heavy posterior updates and optimization based on batched logs.

- Distillation: Distillation: An explicit translation of the offline TS policy into a lightweight representation that enables fast action selection by eliminating the need for online posterior sampling and numerical optimization on low-latency environments.

- Online imitation policy: An explicit policy is trained via Imitation Learning (minimizing KL divergence) to mimic the offline TS policy action distribution for real-time inference.

The authors provide a theoretical analysis decomposing the regret into the standard Bayesian regret plus an imitation error term. They establish general regret bounds comparable to batch TS and generalization bounds for the imitation step. Morevover, they empirically demonstrate that TS imitation policies achieve regret comparable to batch TS ones with significantly lower latency on benchmark datasets and report successful deployment in a large-scale video transcoding system.

**Additional Comments:**

In general the paper is well-written and clear, presenting a complex architectural solution in an understandable manner. While I am not a core domain expert in contextual bandit theory, the analysis appears to be sound and convincing. The required revisions focus on strengthening the scientific rigor of the evidence and clarifying the systems architecture trade-offs.

**Audience:**

Yes

**Audience Explanation:**

This paper bridges the gap between theoretical bandit literature and systems engineering, making it relevant to both ML practitioners and theoreticians. For the first group, the paper addresses a crucial point in industry, i.e., the difficulty of deploying principled exploration algorithms (like TS) on edge devices due to compute/latency constraints. The architectural pattern is highly applicable to recommendation systems and mobile applications. For the second group, the analysis of off-policy regret and the application of localized Rademacher complexity to bound the imitation error contribute to the understanding of how approximation errors propagate in sequential decision-making problems.

**Broader Impact Concerns:**

The authors do not present a specific Broader Impact Statement. I do not see immediate critical ethical concerns requiring a halt to publication.

**Claims And Evidence:**

Yes

**Claims Explanation:**

The claims are generally well-supported by a mix of theoretical proofs and empirical evaluations. In particular, the claim of "order of magnitude reduction in decision-time latency" is substantiated by Table 1, which compares the execution time of the distilled TS against standard TS approaches. Also, the claim that the imitation policy achieves performance "comparable to batch TS" is supported by Figure 2, where the cumulative regret curves of the imitation policies closely track their offline counterparts across multiple datasets.
This claim is supperted also by theoretical analysis regarding regret decomposition and generalization capabilities, which appear mathematically correct based on the derivations provided in the appendices.

**Requested Changes:**

Clarify Offline Computational Cost: The paper emphasizes efficiency and low-latency of online inference, but the distillation step requires running the offline TS policy's inference on N contexts (where N is "orders of magnitude larger" than the batched policy updates T, and can be "in the order of hundred of millions"). The authors should explicitly acknowledge that while online latency is reduced, the offline compute budget increases significantly to generate the synthetic labels. A brief discussion or rough estimate of this offline cost trade-off is necessary to ensure the "efficiency" claim is not misleading, since the massive computational burden is merely shifted from the edge device to the server, rather than being eliminated.

Discuss Model Misspecification: The theoretical guarantees for imitation learning rely on the assumption that the "imitation model class is well-specified". In practice, a simple softmax policy cannot perfectly capture the geometry of a complex Bayesian posterior. The authors should add a remark acknowledging this gap and that irreducible approximation error (bias) exists and how it might impact the convergence rate claim.

More details on RCT results: While impressive, the RCT results are summarized in a way that is not reproducible and cannot be scientifically scrutinized. It is also not clear if we can attribute these results entirely to the new methodology or to exogenous or unmentioned factors.

---

> ### Author Response · Authors · 2026-01-18
> **Author Response**
>
> Dear Reviewer SjvG,
>
> Thank you for your positive assessment of our work and for identifying important points requiring clarification.
>
> 1. Clarifying Offline Computational Cost
> You correctly note that our "efficiency" claim could be misleading since computation is shifted rather than eliminated. We have added discussions that explicitly acknowledge this tradeoff. We have revised our language throughout to emphasize "low online latency" rather than overall efficiency, making clear that the computational burden is shifted to an offline setting where it can leverage powerful backend infrastructure and embarrassingly parallel computation.
>
> 2. Model Misspecification
> We have added discussion in Section 5.2 covering both Bayesian model misspecification and imitation model misspecification. The former is a standard concern for Thompson sampling that our method inherits; the latter is relatively mild given that modern neural networks with sufficient capacity can approximate complex policy distributions. Importantly, the distillation process itself does not amplify misspecification.
>
> 3. RCT Results Detail
> We appreciate this important concern about scientific rigor. We have added a new paragraph in Section 4 providing additional methodological details about our randomized controlled trials, including the control, randomization, scale, and duration. We acknowledge that full reproducibility is limited by proprietary constraints, but we believe these additional details allow readers to better assess the validity of our findings. We have added an anonymized Github repo link in the abstract and our implementation will be released as open-source software upon publication.
>
>
> Thank you for your constructive feedback, which has helped us strengthen the scientific rigor and clarity of our presentation.

---

### Review · Reviewer_Wa3Q · 2025-11-22

**Summary Of Contributions:**

# **Summary of Contributions**

This paper introduces *Distilled Thompson Sampling*, an imitation-learning framework designed to make Thompson Sampling (TS) operationally feasible in large-scale, latency-sensitive contextual bandit settings. The method relocates all posterior inference and optimization steps to an offline environment and learns a parametric policy that imitates TS, enabling fast action selection on resource-constrained devices. The principal contributions are summarized below.

The paper proposes a practical algorithm for offline simulation and imitation of batch TS. By repeatedly updating the posterior on backend servers and generating synthetic TS actions, the method trains an explicit policy representation that can be deployed online without requiring posterior sampling or numerical optimization. This design directly targets system-level constraints such as latency, memory usage, and engineering complexity.

The work demonstrates that the distilled policy can produce actions with extremely low latency—more than an order-of-magnitude faster than conventional TS—while maintaining cumulative regret competitive with batch TS across benchmark tasks, including Wheel, Mushroom, and Warfarin dosage. The approach is validated further through a large-scale real-world application in video transcoding, where the distilled policy improves quality and engagement metrics and is subsequently deployed across major products handling millions of daily video uploads.

The paper also provides theoretical analysis showing that the imitation policy achieves Bayes regret comparable to batch TS, with deviation controlled by per-period imitation error terms that do not accumulate over time. Finally, it establishes generalization guarantees for the imitation objective, showing that abundant unsupervised contexts enable accurate approximation of the TS policy.

# **Key Strengths**

The paper presents a method with significant practical relevance, demonstrated both through controlled experiments and through deployment across multiple production surfaces. The proposed approach produces substantial latency reductions while preserving the performance characteristics of TS. The work offers a rigorous theoretical treatment, including a regret decomposition linking imitation error to Bayes regret, Gaussian process-based analyses, and convergence guarantees for the empirical imitation procedure. The system-level motivation is clear and well-supported, and the contribution addresses real barriers to implementing TS in industrial settings.

# **Key Weaknesses**

The approach inherits the limitations of the underlying TS policy, including sensitivity to prior specification and uncertainty calibration. The theoretical results rely on assumptions such as well-specified Bayesian models, smooth kernels for Gaussian process analysis, and specialized initialization procedures, which may not hold universally. Although offline computation does not affect latency, the posterior-update and TS-simulation phases may be computationally demanding in practice. The off-policy nature of the imitation process introduces analytical complexity and may limit the generality of the theoretical guarantees outside the modeled settings.

**Additional Comments:**

N/A

**Audience:**

Yes

**Audience Explanation:**

The paper tackles a problem that is highly relevant to the TMLR community: enabling principled Bayesian decision-making algorithms to function effectively in large-scale, real-time environments. Thompson Sampling is widely studied in machine learning, reinforcement learning, and online decision-making. However, its direct deployment is often limited by the computational cost of posterior inference and the need for low-latency decisions. The proposed imitation-learning framework preserves the statistical strengths of Thompson Sampling while making it practical for deployment on real systems, including mobile and other resource-limited platforms. This represents a meaningful contribution for both researchers and practitioners.

Several factors support its relevance to TMLR readers:

1. The work provides a bridge between theory and practice by combining regret analysis with empirical validation and deployment results.
2. The imitation-learning view of distilling Bayesian decision policies is general and can be applied to other learning problems, including reinforcement learning and settings with very large action spaces.
3. The paper addresses a real challenge in the deployment of machine learning systems, namely how to operationalize bandit algorithms in environments with strict latency and system constraints.
4. The results have clear implications for ML systems research and contextual bandit methodology, both of which are widely represented in the TMLR audience.

Given these factors, the findings are likely to be of interest to readers working on contextual bandits, Bayesian inference, imitation learning, online learning algorithms, and applied ML systems.

**Broader Impact Concerns:**

The submission focuses on improving the practicality of Thompson Sampling by enabling low latency, large scale deployment through imitation learning. While the method itself is algorithmic and primarily technical in nature, several broader impact considerations arise from its intended use in real world systems.

The first concern involves the deployment of automated decision systems on resource constrained devices, including mobile phones. Such systems can influence user experience, content delivery, and resource allocation at large scale. The paper describes deployment in video upload systems and similar recommendations or ranking environments. These systems may affect user engagement patterns, consumption behavior, or visibility of certain types of content. Although the paper presents positive product level improvements, it would be beneficial to acknowledge that changes in algorithmic policies can have downstream effects on user behavior that are difficult to anticipate.

The second concern relates to fairness and potential disparate impact. Contextual bandit systems deployed across diverse user populations may behave differently for users in regions with limited bandwidth, older devices, or lower quality connectivity. Since the distilled policy is trained to imitate Thompson Sampling under specific contextual conditions, any bias in the underlying reward model, context features, or prior choices will transfer directly to the deployed policy. A brief reflection on fairness considerations, especially given global scale deployment, would strengthen the submission.

A third concern involves privacy and data handling. The method assumes access to large amounts of unsupervised contextual data from user interactions. Although the paper does not manipulate or expose individual level information, the reliance on large internal datasets raises familiar privacy considerations. A short note clarifying that the method operates on aggregated feature representations without requiring additional data collection would help preempt misunderstandings.

Finally, the approach is motivated partly by improvements in engagement metrics, such as watch time. While this is standard in many industry settings, there is a broader societal discussion on how engagement optimizing systems may contribute to addictive usage patterns. Although this paper does not introduce new mechanisms that amplify such effects, a brief acknowledgment that better system performance may influence user behavior would make the broader impact statement more complete.

Overall, the work does not introduce direct ethical risks, but the scale and nature of the deployment environment warrants a short discussion of fairness, privacy, and behavioral effects.

**Claims And Evidence:**

Yes

**Claims Explanation:**

The submission’s main empirical and theoretical claims are supported by evidence that is, in general, accurate, well-structured, and appropriate for the stated contributions. The empirical evaluation spans both standard contextual bandit benchmarks and a large-scale real-world application involving video transcoding. Across all synthetic and public datasets, the results consistently show that the distilled imitation policies match the cumulative regret of their corresponding Thompson Sampling variants while substantially reducing latency. The experimental design, including batching, hyperparameter choices, and evaluation metrics, is clearly described and aligns with precedents established in the literature.

The large-scale deployment evidence is particularly compelling. The authors describe improvements observed in internal randomized controlled trials, including increases in video quality, watch time, and user engagement across multiple platforms. While proprietary and therefore not independently reproducible, the details provided are sufficient to substantiate the practical relevance of the method and are consistent with standard reporting practices for industrial systems research.

The theoretical claims are supported by rigorous derivations and proofs. The regret decomposition is clearly presented, and the argument linking imitation accuracy to Bayes regret is logically sound and consistent with existing analyses of Thompson Sampling. The assumptions underlying the theoretical results are explicitly stated, and although some are strong (e.g., Gaussian process models, smooth kernels, availability of abundant unsupervised contexts), the theoretical contributions themselves are internally coherent and mathematically correct within that framework.

Taken together, the combination of thorough empirical evaluation, carefully articulated deployment evidence, and detailed theoretical analysis provides convincing support for the claims made in the submission.

**Requested Changes:**

# **Critical**

1. Clarify the assumptions used in the theoretical analysis: The regret bounds depend on specific modeling assumptions, including Gaussian process priors, smoothness of kernels, and a particular initialization procedure. A clearer explanation of when these assumptions hold in realistic settings, and how restrictive they are, would improve the transparency of the theoretical results.

2. Provide more detailed intuition for the off policy analysis: The paper shows that imitation error does not accumulate over time, but the surrounding explanation is difficult to follow for readers who are not already familiar with this line of work. Additional intuition about why off policy imitation remains stable would be beneficial.

3. Expand the discussion of offline computational cost: The method moves all posterior inference and optimization to an offline stage, but the submission provides limited detail about the cost of this process in practice. Since practical deployability is a central claim, a clearer description of the computational requirements and update frequency is necessary.

# **Non Critical**

1. Add a short discussion on model misspecification: Since the distilled policy inherits the behavior of the Thompson Sampling model, any misspecification in the Bayesian model can also affect the imitation learner. A brief discussion of this issue would strengthen the submission.

2. Clarify the extent to which the method may generalize beyond contextual bandits: The paper mentions potential relevance to reinforcement learning, but it would be helpful to outline which elements of the current framework do or do not translate to sequential decision settings.

3. Provide a concise summary of hyperparameters used for the imitation model: A small table or appendix listing neural architectures, optimizer settings, and training parameters would improve reproducibility.

4. Offer additional qualitative detail for the video transcoding application: While proprietary constraints are understood, a clearer description of the structure of contexts, actions, and rewards in this setting would help readers appreciate the deployment environment.

5. Improve the readability of some proofs: Several technical results, including those involving mutual information and regret decomposition, are presented in a dense manner. Minor editorial restructuring could make the proofs easier to follow.

6. Discuss the behavior of the algorithm under irregular batch sizes: The framework assumes that batching is feasible and relatively stable. A short comment on how performance might be affected by highly variable batch sizes would add useful context.

---

> ### Author Response · Authors · 2026-01-18
> **Author Response**
>
> Dear Reviewer Wa3Q,
>
> Thank you for your comprehensive and thoughtful review. We address your concerns below.
>
> 1. Clarifying Theoretical Assumptions
> We have added a new subsection "Discussion of Assumptions" that explicitly addresses when our theoretical assumptions hold in practice, including discussion of the Gaussian process assumption, smoothness conditions, and the initialization scheme.
>
> 2. Intuition for Off-Policy Analysis
> We have expanded the discussion at the end of Section 5.1 to provide clearer intuition. The key insight is that under Bayes regret (averaging over the prior), whether the history was formed from on-policy or off-policy data does not directly affect the regret decomposition. Thompson sampling's optimality property (selecting actions according to their posterior probability of being optimal) holds regardless of how the posterior was formed. As a result, each period's imitation error contributes additively rather than multiplicatively, preventing the cascade of errors that a pessimistic analysis might suggest.
>
> 3. Offline Computational Cost
> We have expanded our discussion of the online-offline computational tradeoff. We now explicitly describe the typical scale of N, computation time (minutes with modern distributed infrastructure), and the embarrassingly parallel nature of the offline computation in Section 7. We emphasize that this tradeoff is most favorable when offline computation can leverage powerful backend servers while online computation must execute on resource-constrained mobile devices.
>
> 4. Model Misspecification
> We have added discussion in Section 5.2 addressing both Bayesian model misspecification and imitation model misspecification. Importantly, the distillation process does not amplify misspecification—our approach inherits limitations of the underlying Thompson sampling policy but does not introduce additional sources of error. Thank you for pushing us to candidly discuss the limitations of our approach.
>
> 5. Generalization Beyond Contextual Bandits
> We have added a paragraph in Section 7 discussing potential extensions to reinforcement learning, including behavioral cloning with trajectory-level imitation, inverse RL approaches, and generative adversarial imitation learning.
>
> 7. Experimental Details
> We have expanded the dataset description to provide more detail on the structure of contexts (video features, network features, device features, user features) in our video transcoding application. Please note that we summarize further details such as hyperparameters in Appendix E. We have also added an anonymized Github repo link in the abstract and our implementation will be released as open-source software upon publication.
>
> 8. Broader Impact Concerns
> We have added a comprehensive Broader Impact Statement.
>
> Thank you for pushing us to make these aspects of our work more explicit and rigorous.

---

### Review · Reviewer_Gu4r · 2026-01-12

**Summary Of Contributions:**

This paper proposes an imitation-learning-based algorithm that distills a Thompson sampling (TS) policy into a lightweight neural network model, enabling efficient online decision making in mobile devices. The TS policy is updated using batched data collected under the imitation policy in an offline manner. The experimental evaluation demonstrates that the proposed method achieves comparable performance to batch TS and reduces the decision-making latency. In addition, the proposed method successfully deploys in the multiple video upload system for Meta, demonstrating its real-world applicability and impact. Furthermore, the paper presents the theoretical analyses that justify the effectiveness of the proposed algorithm on large-scale services.

[Strengths]
- This paper proposes a simple yet effective TS algorithm for efficient online decision making in mobile devices.
- The effectiveness of the proposed algorithm is demonstrated through empirical evaluation on both synthetic and real-world problems.
- The theoretical analysis justifying the proposed algorithm is also provided.

[Weaknesses]
I do not have any critical weaknesses to point out.

**Audience:**

Yes

**Audience Explanation:**

The proposed algorithm is relatively simple but powerful in real-world applications, and the theoretical analysis is also provided. Therefore, I believe that researchers and practitioners in machine learning and related fields would be interested in the findings of this paper.

**Broader Impact Concerns:**

Although I do not have any critical concerns regarding the ethical implications of this work, it might be nice to include a Broader Impact Statement to discuss potential effects caused by operating large-scale systems, such as a video upload system, using the proposed algorithm, if possible.

**Claims And Evidence:**

Yes

**Claims Explanation:**

The effectiveness of the proposed method is well demonstrated through empirical evaluation. Additionally, the theoretical analyses justifying the proposed algorithm are provided. The paper is generally well-written.

**Requested Changes:**

I do not have any critical changes to suggest. However, I have a few minor comments below.

- In Figure 2 (d), the label of the vertical axis should be "Running Average of Rewards."

- It might be better to add a brief explanation about the structure of this paper in the Introduction. Since this paper includes both empirical evaluation and theoretical analysis of the proposed algorithm, it would be helpful for readers to understand the organization of this paper.

- In Table 2, the latency of the decision-making on Intel Xeon E5-2680, which is not a mobile device. It would be great if the authors report or discuss the latency on actual mobile devices. However, this is not a strict requirement.

- On the last in the second paragraph of page 17, "...entire model space $m \in \mathcal{M}$," should be "...entire model space $m \in \mathcal{M}$." (The comma should be a period.)

- I could not find the supplementary material, although the authors mentioned, "We include our open-sourced implementation as supplementary materials." I would be great if the authors could share the link to the code repository after acceptance.

---

> ### Author Response · Authors · 2026-01-18
> **Author Response**
>
> Dear Reviewer Gu4r,
>
> Thank you for your positive evaluation of our work and for the constructive suggestions.
>
> - Figure 2(d) Label: Thank you for catching this. We have corrected the y-axis label.
> - Paper Structure in Introduction: We have added a paragraph at the end of the Introduction providing a roadmap of the paper's organization.
> - Latency on Mobile Devices: This is an excellent point. Our Table 1 measurements use server hardware for controlled comparison across methods. We have added a new paragraph "Mobile Latency" in Section 4 discussing mobile device performance.
> - Code Release: We have added an anonymized Github repo link in the abstract and our implementation will be released as open-source software upon publication.
> - Broader Impact Statement: We have added a dedicated Broader Impact Statement at the end of Section 7. We appreciate this suggestion to make broader considerations explicit.

---

> > ### Comment · Reviewer_Gu4r · 2026-01-22
> >
> > Thank you for the reply and for revising the paper. I have satisfied the revised paper.
> > I have no more concerns for this paper.
> >
> > Best,

---

### Decision · Action_Editor_wDqZ · 2026-02-20

**Recommendation:** Accept as is

**Additional Comments:**

This paper proposes distilled Thompson sampling. The key idea is to offload costly posterior updates to servers and deploy a distilled policy to mobile devices. This results in a much lower inference time. In practice, a quick and good recommendation on a mobile device can make a huge difference in how engaged the user is. The proposed approach is analyzed and evaluated empirically on classic bandit benchmarks. It is also applied to video transcoding, where the distilled policy improves engagement metrics and is then deployed across major products at Meta.

This paper received three positive reviews. The reviewers had only a few major requests:

* Clarify offline computational cost

* Discuss sensitivity to prior

* Discuss model misspecification

* Clarify assumptions in the analysis and provide intuition for the off-policy analysis

* Discuss ethical concerns

The authors addressed these and I strongly support acceptance of this paper.

**Audience:**

Yes

**Audience Explanation:**

This paper is on the intersection of classic statistics, systems, and modern machine learning; and each community can learn something from it. Great work!

**Claims And Evidence:**

Yes

**Claims Explanation:**

This paper aims to improve the inference-time cost of Thompson sampling and reports more than an order-of-magnitude improvement in experiments. Experiments on classic bandit benchmarks are complemented by a real-world deployment of the method. The approach is analyzed and the theoretical claims are backed by proofs.